# Generalization of Model-Agnostic Meta-Learning Algorithms: Recurring and Unseen Tasks

**Alireza Fallah**
EECS Department
Massachusetts Institute of Technology
afallah@mit.edu

**Aryan Mokhtari**
ECE Department
The University of Texas at Austin
mokhtari@austin.utexas.edu

**Asuman Ozdaglar**
EECS Department
Massachusetts Institute of Technology
asuman@mit.edu

## Abstract

In this paper, we study the generalization properties of Model-Agnostic Meta-Learning (MAML) algorithms for supervised learning problems. We focus on the setting in which we train the MAML model over $m$ tasks, each with $n$ data points, and characterize its generalization error from two points of view: First, we assume the new task at test time is one of the training tasks, and we show that, for strongly convex objective functions, the expected excess population loss is bounded by $\mathcal{O}(1/mn)$. Second, we consider the MAML algorithm's generalization to an unseen task and show that the resulting generalization error depends on the total variation distance between the underlying distributions of the new task and the tasks observed during the training process. Our proof techniques rely on the connections between algorithmic stability and generalization bounds of algorithms. In particular, we propose a new definition of stability for meta-learning algorithms, which allows us to capture the role of both the number of tasks $m$ and number of samples per task $n$ on the generalization error of MAML.

## 1 Introduction

In several machine learning problems, it is of interest to design algorithms that can be adjusted based on previous experiences and tasks to perform better on a new task. In particular, meta-learning algorithms achieve such a goal through various approaches, including finding a proper meta-initialization for the new task [1–3], updating the model architecture [4–6], or learning the parameters of optimization algorithms [7, 8].

A popular meta-learning framework that has shown promise in practice is Model-Agnostic Meta-Learning (MAML), which was first introduced in [1]. MAML algorithm uses available training data on a number of tasks to come up with a meta-initialization that performs well after it is slightly updated at test time with respect to the new task. In other words, unlike standard supervised learning, in which we aim to find a model that generalize well to a new task *without any adaptation step*, in MAML our goal is to find an initial model for learning a new task when *we have access to limited labeled data for that task* to run one (or a few) step(s) of stochastic gradient descent (SGD).

As shown in Fig. 1, in MAML we are given $m$ tasks with $m$ corresponding datasets $\{\mathcal{S}_i\}_{i=1}^m$ in the training phase. Once the model is trained ($w_{\text{train}}^*$), a new task is revealed at test time for which we have access to $K$ *labeled* samples drawn from $\mathcal{D}_{\text{test}}$. We use these labeled samples of the new task to

update the trained model by running a step of SGD leading to a new model for the test task ($w_{\text{new}}^*$). We finally evaluate the performance of the updated model over the test task, denoted by $\mathcal{L}_{test}(w_{\text{new}}^*)$.

MAML and its variants have been extensively studied over the past few years from both empirical and theoretical point of view [2, 9–16]. In particular, [13] provided convergence guarantees for MAML algorithm under the assumption that access to fresh samples at any round of the training stage is possible, and [15] extended this results to the case that multiple gradient steps can be performed at test time. However, one shortcoming of such analysis is that, at training stage, we often do not have access to fresh samples at every iteration. Instead, we have access to a large set of realized samples and we typically do multiple passes over the data points during the training stage.

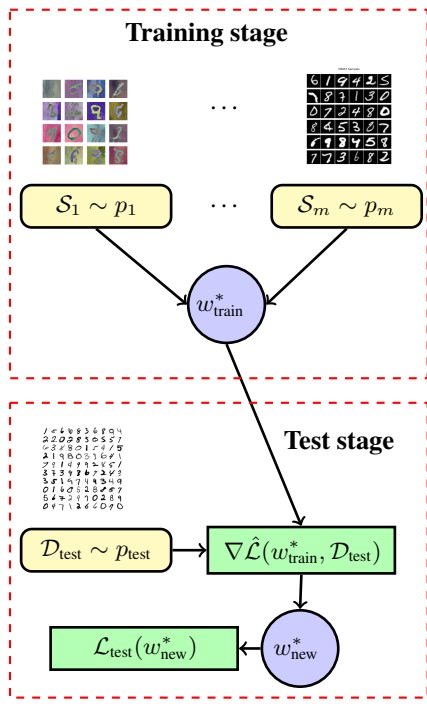

Figure 1: MAML framework

Hence, it is essential to come up with a novel analysis that addresses this issue by characterizing the training error and generalization error of MAML separately. In this paper, we accomplish this goal and showcase the role of different problem parameters in the generalization error of MAML. Specifically, we assume that we are given $m$ supervised learning tasks, with (possibly different) underlying distributions $p_1, \ldots p_m$, where for each task we have access to $n$ samples[1]. As we measure the performance of a model by its loss after one step of SGD adaptation with $K$ samples, the problem that one can solve in the training phase is minimizing the average loss, over all given $m$ tasks and their $n$ samples, after one step of SGD with $K$ samples. This empirical loss can be considered as a surrogate for the desired expected loss (with respect to tasks data) over all $m$ tasks. Here, we focus on the case that MAML is used to solve this empirical minimization problem, and our goal is to quantify the test error of MAML output. To tackle this problem, we first briefly revisit the results from the optimization literature to bound the training error of MAML, assuming that the loss functions are strongly convex. We next turn to the main focus of our paper which is the generalization properties of MAML. More specifically, we address the following questions:

- *If one of the $m$ given tasks recurs uniformly at random at test time, then how well (in expectation) would the trained model perform after adaptation with SGD over the fresh samples of that task?* In other words, having training error minimized, what would be the generalization error and our guarantee on test error? Here, we show that for strongly convex objective functions, we could achieve a generalization error that decays at $\mathcal{O}(1/mn)$. Our analysis builds on the connections between algorithmic stability and generalization of the output of algorithms. While this relation is well-understood in classic statistical learning [17, 18], here we propose a novel stability definition for meta-learning algorithms which allows us to restore such connection for our setting.

- *Assuming that the task at test time is NOT one of the $m$ tasks at training, how would the model perform on that task after the adaptation step?* We answer this question by focusing on the case that the revealed task at the test time is a new *unseen* task with underlying data distribution $p_{m+1}$, and formally characterizing the generalization error of MAML in this case. We show that when the task at test time is new, the generalization error also depends on the total variation distance between $p_{m+1}$ and $p_1, \ldots, p_m$.

**Related work:** Recently, there has been significant progress in studying theoretical aspects of meta-learning, in particular, MAML. Authors in [19] proposed iMAML which updates the model using an approximation of one step of proximal point method and studied its convergence. In [20], authors introduced the task-robust MAML by considering a minimax formulation rather than minimization. Several papers have also studied MAML through more general frameworks such as bilevel optimization [21], stochastic compositional optimization [22], and conditional stochastic

---

[1]More precisely, in our analysis we take $2n$ samples per each task to simplify derivations.

optimization [23]. Also, several works have studied the extension of meta-learning theory to online learning [24, 3], federated learning [25], and reinforcement learning [26, 27].

The most relevant paper to our work is [28] that studies generalization of meta-learning algorithms using stability techniques and shows a $\mathcal{O}(1/\sqrt{m})$ bound for nonconvex loss functions. Here we focus on strongly convex objective functions and present an analysis that differs from this work in two fundamental aspects. First, we present a different notion of stability that allows us to capture the number of data points per task in our bound. In particular, our stability notion measures sensitivity of the algorithm to perturbations that involve changing $K$ data points which is the data unit involved in the adaptation step of the MAML algorithm. This enables us to obtain a much tighter bound $\mathcal{O}(1/mn)$ (compared to $\mathcal{O}(1/m)$ achieved in [28] for strongly convex functions), highlighting the dependence on the number of the data samples available for each task. Second, we also consider the generalization of MAML for the case that the task at test time is not one of the available tasks during the training stage.

The generalization of MAML has also been studied in [29] from an empirical point of view. In particular, they show that the generalization of MAML to new tasks is correlated with the coherence between their adaptation trajectories in parameter space. This is aligned with the connection of generalization and closeness of underlying distributions that we observe in our results.

## 2  Problem formulation

In this paper, we consider the supervised learning setting, where each data point is denoted by $z = (x, y) \in \mathcal{Z}$ with $x \in \mathcal{X}$ being the input (feature vector) and $y \in \mathcal{Y}$ being its corresponding label. We use the loss function $l : \mathbb{R}^d \times \mathcal{Z} \to \mathbb{R}^+$ to evaluate the performance of a model parameterized by $w \in \mathcal{W}$, where $\mathcal{W}$ is a convex and closed subset of $\mathbb{R}^d$. In other words, for a data point $z = (x, y) \in \mathcal{Z}$, the loss $\ell(w, z)$ denotes the error of model $w$ in predicting the label $y$ given input $x$.

We consider access to $m$ tasks denoted by $\mathcal{T}_1, \ldots, \mathcal{T}_m$, where the data corresponding to each task $\mathcal{T}_i$ is generated from a distinct distribution $p_i$. The *population loss* corresponding to task $\mathcal{T}_i$ for model $w$ is defined as $\mathcal{L}_i(w) := \mathbb{E}_{z \sim p_i}[\ell(w, z)]$.

We further use the notation $\hat{\mathcal{L}}(w; \mathcal{D})$ to denote the *empirical loss* corresponding to dataset $\mathcal{D}$, which is defined as the average loss of $w$ over the samples of dataset $\mathcal{D}$, i.e., $\hat{\mathcal{L}}(w; \mathcal{D}) := \frac{1}{|\mathcal{D}|} \sum_{z \in \mathcal{D}} \ell(w, z)$, where $|\mathcal{D}|$ is the size of dataset $\mathcal{D}$. In general, and throughout the paper, we use the *hat notation* to distinguish *empirical* losses from population losses.

Our goal is to find $w \in \mathcal{W}$ that performs well on average[2] over all tasks, after it is updated with respect to the new task and by using one step of stochastic gradient descent (SGD) with a batch of size $K$. To formally introduce this problem we first define the function $F_i(w)$ which captures the performance of model $w$ over task $\mathcal{T}_i$ once it is updated by a single step of SGD,

$$F_i(w) := \mathbb{E}_{\mathcal{D}_i^{\text{test}}} \left[ \mathcal{L}_i \left( w - \alpha \nabla \hat{\mathcal{L}}(w, \mathcal{D}_i^{\text{test}}) \right) \right] = \mathbb{E}_{\mathcal{D}_i^{\text{test}}} \mathbb{E}_{z \sim p_i} \left[ \ell \left( w - \frac{\alpha}{K} \sum_{z' \in \mathcal{D}_i^{test}} \nabla \ell(w, z'), z \right) \right] \quad (1)$$

where $\mathcal{D}_i^{\text{test}}$ is a batch with $K$ different samples, drawn from the probability distribution $p_i$. Note that the outer expectation is taken with respect to the choice of elements of $\mathcal{D}_i^{\text{test}}$ while the inner one is taken with respect to the data of task $i$.

As our goal is to find a model that performs well after one step of adaptation over all $m$ tasks, we minimize the average expected loss over all given tasks, which can be written as

$$\min_{w \in \mathcal{W}} F(w) := \frac{1}{m} \sum_{i=1}^{m} F_i(w). \quad (2)$$

As the underlying distribution of tasks are often unknown in most applications, we are often unable to directly solve the problem in (2). On the other hand, for each task, we often have access to data points that are drawn according to their data distributions. Therefore, instead of solving (2), we solve its sample average surrogate problem in which each $F_i$ is approximated by its empirical loss.

---

[2]Our analysis can be extended to the case that the distribution over tasks is not uniform.

To formally define the empirical loss for each task, suppose for each task $\mathcal{T}_i$ we have access to a training set $\mathcal{S}_i$, where its elements are drawn independently according to the probability distribution $p_i$. We further divide the set $\mathcal{S}_i$ into two disjoint sets of size $n$ defined as $\mathcal{S}_i^{\text{in}}$ and $\mathcal{S}_i^{\text{out}}$, i.e., $\mathcal{S}_i := \{\mathcal{S}_i^{\text{in}}, \mathcal{S}_i^{\text{out}}\}$ and $|\mathcal{S}_i^{\text{in}}| = |\mathcal{S}_i^{\text{out}}| = n$. Here, we use the elements of the $\mathcal{S}_i^{\text{in}}$ to estimate the inner gradient $\nabla \hat{\mathcal{L}}(w, \mathcal{D}_i^{\text{test}})$ and use the samples in the set $\mathcal{S}_i^{\text{out}}$ to estimate the outer function $\mathcal{L}_i(.)$. Specifically, we define the sample average of $F_i$ using data sets $\mathcal{S}_i^{\text{in}}$ and $\mathcal{S}_i^{\text{out}}$ as

$$
\hat{F}_i(w, \mathcal{S}_i) := \frac{1}{\binom{n}{K}} \sum_{\mathcal{D}_i^{\text{in}} \subset \mathcal{S}_i^{\text{in}} | \mathcal{D}_i^{\text{in}} | = K} \hat{\mathcal{L}}\left(w - \alpha \nabla \hat{\mathcal{L}}(w, \mathcal{D}_i^{\text{in}}), \mathcal{S}_i^{\text{out}}\right) \tag{3}
$$

$$
= \frac{1}{\binom{n}{K}} \sum_{\mathcal{D}_i^{\text{in}} \subset \mathcal{S}_i^{\text{in}} | \mathcal{D}_i^{\text{in}} | = K} \frac{1}{n} \sum_{z \in \mathcal{S}_i^{\text{out}}} \ell\left(w - \frac{\alpha}{K} \sum_{z' \in \mathcal{D}_i^{\text{in}}} \nabla \ell(w, z'), z\right).
$$

This expression shows that we use all $n$ elements of $\mathcal{S}_i^{\text{out}}$ to approximate the expectation required for the computation of $\mathcal{L}_i$, and we approximate the expectation with respect to the test set by averaging over all subsets of $\mathcal{S}_i^{\text{in}}$ that have $K$ elements. Given this expression, the sample average approximation (empirical loss) of Problem (1) is given by

$$
\arg\min_{w \in \mathcal{W}} \hat{F}(w, \mathcal{S}) := \frac{1}{m} \sum_{i=1}^{m} \hat{F}_i(w, \mathcal{S}_i), \tag{4}
$$

where $\mathcal{S} := \{\mathcal{S}_i\}_{i=1}^{m}$ is defined as the concatenation of all tasks data sets.

Having the dataset $\mathcal{S}$, a (possibly randomized) optimization algorithm $\mathcal{A}$ with output $\mathcal{A}(\mathcal{S})$ can be used to find an approximate solution to the problem in (4). The error of this solution with respect to the MAML empirical loss, i.e., $\hat{F}(\mathcal{A}(\mathcal{S}), \mathcal{S}) - \min_{\mathcal{W}} \hat{F}(., \mathcal{S})$, is called *training error*. In this paper, we are mainly interested to bound the *test error* which is the error of $\mathcal{A}(\mathcal{S})$ with respect to the population loss, i.e., $F(\mathcal{A}(\mathcal{S})) - \min_{\mathcal{W}} F$. The test error is also sometimes called *excess (population) loss*. Note that the expected test error can be decomposed into three terms:

$$
\mathbb{E}_{\mathcal{A}, \mathcal{S}}\left[F(\mathcal{A}(\mathcal{S})) - \min_{\mathcal{W}} F\right] \quad \text{(test error)} =
$$

$$
\underbrace{\mathbb{E}_{\mathcal{A}, \mathcal{S}}\left[F(\mathcal{A}(\mathcal{S})) - \hat{F}(\mathcal{A}(\mathcal{S}), \mathcal{S})\right]}_{\text{generalization error}} + \underbrace{\mathbb{E}_{\mathcal{A}, \mathcal{S}}\left[\hat{F}(\mathcal{A}(\mathcal{S}), \mathcal{S}) - \min_{\mathcal{W}} \hat{F}(., \mathcal{S})\right]}_{\text{training error}} + \underbrace{\mathbb{E}_{\mathcal{S}}\left[\min_{\mathcal{W}} \hat{F}(., \mathcal{S})\right] - \min_{\mathcal{W}} F}_{\leq 0}.
$$

It can be verified that the expectation of the third term (over $\mathcal{A}$ and $\mathcal{S}$) is non-positive since $\mathbb{E}_{\mathcal{S}}[\min_{\mathcal{W}} \hat{F}(., \mathcal{S})] \leq \min_{\mathcal{W}} \mathbb{E}_{\mathcal{S}}[\hat{F}(., \mathcal{S})]$ and $\mathbb{E}_{\mathcal{S}}[\hat{F}(., \mathcal{S})] = F$. Hence, to bound the expected test error, we should bound the expectation of training and generalization errors.

The Model-Agnostic Meta-Learning (MAML) method proposed in [1] is designed to solve the empirical minimization problem defined in (4). The steps of MAML are outlined in Algorithm 1. MAML solves Problem (4) by using SGD update for the average loss function $\hat{F}(w, \mathcal{S})$. To better highlight this point, note that the gradient of $\nabla \hat{F}(w, \mathcal{S})$ can be written as $\frac{1}{m} \sum_{i=1}^{m} \nabla \hat{F}_i(w, \mathcal{S}_i)$, where the $i$-th term corresponding to task $\mathcal{T}_i$ is given by

$$
\nabla \hat{F}_i(w, \mathcal{S}_i) = \frac{1}{\binom{n}{K}} \sum_{\substack{\mathcal{D}_i^{\text{in}} \subset \mathcal{S}_i^{\text{in}} \\ |\mathcal{D}_i^{\text{in}}| = K}} \left[ (I_d - \alpha \nabla^2 \hat{\mathcal{L}}(w, \mathcal{D}_i^{\text{in}})) \times \nabla \hat{\mathcal{L}}\left(w - \alpha \nabla \hat{\mathcal{L}}(w, \mathcal{D}_i^{\text{in}}), \mathcal{S}_i^{\text{out}}\right) \right], \tag{5}
$$

which involves the second-order information of the loss function. Therefore, to compute a mini-batch approximation for the above gradient, we consider the batches $\mathcal{D}_i^{\text{in}} \subset \mathcal{S}_i^{\text{in}}$ with size $K$ and $\mathcal{D}_i^{\text{out}} \subset \mathcal{S}_i^{\text{out}}$ with $b$ elements. Replacing the above sums with their batch approximations leads to the following stochastic gradient approximation

$$
g_i(w; \mathcal{D}_i^{\text{in}}, \mathcal{D}_i^{\text{out}}) := (I_d - \alpha \nabla^2 \hat{\mathcal{L}}(w, \mathcal{D}_i^{\text{in}})) \nabla \hat{\mathcal{L}}\left(w - \alpha \nabla \hat{\mathcal{L}}(w, \mathcal{D}_i^{\text{in}}), \mathcal{D}_i^{\text{out}}\right), \tag{6}
$$

which is indeed an unbiased estimator of the gradient $\nabla \hat{F}_i(w, \mathcal{S}_i)$ in (5). If for each task we perform the update of SGD with $g_i$ and then compute their average it would be similar to running SGD for

---
**Algorithm 1:** MAML [1]
---
**Input:** The set of datasets $\mathcal{S} = \{\mathcal{S}_i\}_{i=1}^m$ with $\mathcal{S}_i = \{\mathcal{S}_i^{\text{in}}, \mathcal{S}_i^{\text{out}}\}$; test time batch size $K$; # of tasks summoned at each round $r$; # of iterations $T$.
Choose arbitrary initial point $w^0 \in \mathcal{W}$;
**for** $t = 0$ to $T - 1$ **do**
    Choose $r$ tasks uniformly at random (out of $m$ tasks) and store their indices in $\mathcal{B}_t$;
    **for** all $\mathcal{T}_i$ with $i \in \mathcal{B}_t$ **do**
        Sample a batch $\mathcal{D}_i^{t,\text{in}}$ of $K$ different elements from $\mathcal{S}_i^{\text{in}}$ with replacement;
        Sample a batch $\mathcal{D}_i^{t,\text{out}}$ of size $b$ from $\mathcal{S}_i^{\text{out}}$ and with replacement;
        $w_i^{t+1} := w^t - \beta_t \left( I_d - \alpha \nabla^2 \hat{\mathcal{L}}(w^t, \mathcal{D}_i^{t,\text{in}}) \right) \nabla \hat{\mathcal{L}} \left( w^t - \alpha \nabla \hat{\mathcal{L}}(w^t, \mathcal{D}_i^{t,\text{in}}), \mathcal{D}_i^{t,\text{out}} \right);$
    **end for**
    $w^{t+1} := r_{\mathcal{W}} \left( \frac{1}{r} \sum_{i \in \mathcal{B}_t} w_i^{t+1} \right);$
**end for**
**Return:** $w^T$ and $\bar{w}^T := \frac{1}{T+1} \sum_{t=0}^{T} w^t$
---

the average loss $\nabla \hat{F}(w, \mathcal{S})$. This is exactly how MAML is implemented in practice as outlined in Algorithm 1. In this paper, we consider a constrained problem, and as a result, we also need an extra projection step in the last step to ensure the feasibility of iterates. Finally, the output of MAML could be the last iterate $w^T$ or the time-average of all iterates $\bar{w}^T := \frac{1}{T+1} \sum_{t=0}^{T} w^t$.

As stated earlier, the convergence properties of MAML-type methods from an optimization point of view have been studied recently under different set of assumptions. In this paper, as we characterize the sum of training error and generalization error, we briefly discuss the optimization error of MAML when it is used to solve the empirical problem in (4). However, the main focus of this paper is on studying the generalization error of MAML with respect to new samples and new tasks. Specifically, we aim to address the following questions: **(i)** How well does the solution of (4) *generalize* to the main problem of interest in (2)? This could be seen as the *generalization error* of the MAML algorithm over new samples for recurring tasks. **(ii)** How well does the solution of (4) *generalize* to samples from new unseen tasks? To be more precise, how would the obtained model preform if the new task is not one of the $m$ tasks $\mathcal{T}_1, \ldots, \mathcal{T}_m$ observed at training, and it is rather a new, *unseen task* $\mathcal{T}_{m+1}$ with an unknown underlying distribution $p_{m+1}$? In the upcoming sections, we answer these questions on the generalization properties of MAML in detail and characterize the role of number of tasks $m$, number of samples per task $n$, and number of labeled samples revealed at test time $K$.

## 3 Theoretical results

In this section, we formally characterize the excess population loss (test error) of the MAML solution, when we measure the performance of a model after one step of SGD adaptation. In particular, we first discuss the training error of MAML in detail. Then, we establish a generalization error bound for the case that the solution of MAML is evaluated over new samples of a recurring task. Finally, we state the generalization error of MAML once its solution is applied to a new unseen task. Before stating our results, we mention our required assumptions.

**Assumption 1.** *For any $z \in \mathcal{Z}$, the function $\ell(., z)$ is twice continuously differentiable. Furthermore, we assume it satisfies the following properties for any $w, u \in \mathbb{R}^d$:*

*(i) For any $z \in \mathcal{Z}$, the function $\ell(., z)$ is $\mu$-strongly convex, i.e., $\|\nabla \ell(w, z) - \nabla \ell(u, z)\| \geq \mu \|w - u\|$;*

*(ii) The gradient norm is uniformly bounded by $G$ over $\mathcal{W}$, i.e., $\|\nabla \ell(w, z)\| \leq G$;*

*(iii) The loss is $L$-smooth over $\mathbb{R}^d$, i.e., $\|\nabla \ell(w, z) - \nabla \ell(u, z)\| \leq L \|w - u\|$;*

*(iv) Hessian is $\rho$-Lipschitz continuous over $\mathbb{R}^d$, i.e., $\|\nabla^2 \ell(w, z) - \nabla^2 \ell(u, z)\| \leq \rho \|w - u\|$.*

We also require the following assumption on the tasks distribution. This assumption implies that, with probability one, a set of finite samples generated from a distribution $p_i$ are all different.

**Assumption 2.** *We assume $\mathcal{Z}$ is a Polish space (i.e., complete, separable, and metric) and $\mathcal{F}_{\mathcal{Z}}$ is the Borel $\sigma$-algebra over $\mathcal{Z}$. Moreover, for any $i$, $p_i$ is a non-atomic probability distribution over $(\mathcal{Z}, \mathcal{F}_{\mathcal{Z}})$, i.e., $p_i(z) = 0$ for every $z \in \mathcal{Z}$.*

### 3.1 Training error

While the main focus of this paper is on studying the population error of MAML algorithm, we first study its training error which is required to provide characterization of the excess loss of MAML. To do so, we first state the following result from [24] and [13] on the strong convexity and smoothness of $\ell(w - \alpha\nabla\hat{\mathcal{L}}(w, \mathcal{D}), z)$ for any batch $\mathcal{D}$ and any $z \in \mathcal{Z}$.

**Lemma 1** ([13] & [24]). *If Assumption 1 holds, then for an arbitrary batch $\mathcal{D}$ and $z \in \mathcal{Z}$, and with $\alpha \leq \frac{1}{L}$, the function $\ell(w - \alpha\nabla\hat{\mathcal{L}}(w, \mathcal{D}), z)$ is $4L + 2\alpha\rho G$ smooth over $\mathcal{W}$. Furthermore, $\ell(w - \alpha\nabla\hat{\mathcal{L}}(w, \mathcal{D}), z)$ is $\frac{\mu}{8}$-strongly convex, if $\alpha \leq \min\{\frac{1}{2L}, \frac{\mu}{8\rho G}\}$.*

An immediate consequence of this Lemma is that the MAML empirical loss $\hat{F}$ defined in (4) is also $\mu/8$-strongly convex and $4L + 2\alpha\rho G$ smooth over $\mathcal{W}$. In addition, it can be shown that the norm of $g_i(w; \mathcal{D}_i^{\text{in}}, \mathcal{D}_i^{\text{out}})$ defined in (6), which is the unbiased gradient estimate used in MAML, is uniformly bounded above; for more details check Lemma 5 in Appendix A. Having these properties of the MAML empirical loss established, we next state the following proposition on the training error of MAML. This result is obtained by slightly modifying the well-known results on the convergence of SGD in [30–32] in order to take into account the stepsize constraints that are imposed by generalization analysis. For completeness, the proof of this result is provided in Appendix B.

**Proposition 1.** *Consider $\hat{F}(., \mathcal{S})$ defined in (4) with $\alpha \leq \min\{\frac{1}{2L}, \frac{\mu}{8\rho G}\}$. If Assumption 1 holds, then for MAML with $\beta_t = \min(\beta, \frac{8}{\mu(t+1)})$ for $\beta \leq 8/\mu$, and for any set $\mathcal{S}$, the last iterate $w^T$ satisfies*

$$\mathbb{E}\left[\hat{F}(w^T, \mathcal{S}) - \hat{F}(w_{\mathcal{S}}^*, \mathcal{S})\right] \leq \mathcal{O}(1)\frac{G^2(1 + \frac{1}{\beta\mu})}{\mu^2}\left(\frac{L + \rho\alpha G}{T} + \frac{G}{\sqrt{T}}\right), \qquad (7)$$

*and the time-average of iterates $\bar{w}^T$ satisfies*

$$\mathbb{E}\left[\hat{F}(\bar{w}^T, \mathcal{S}) - \hat{F}(w_{\mathcal{S}}^*, \mathcal{S})\right] \leq \mathcal{O}(1)\frac{G^2(\log(T) + \frac{1}{\beta\mu})}{\mu T},$$

*where $w_{\mathcal{S}}^* := \arg\min_{w \in \mathcal{W}} \hat{F}(., \mathcal{S})$ and the expectations are taken over the randomness of algorithm.*

In the above expressions, the notation $\mathcal{O}(1)$ only hides absolute constants. It is worth noting that the term $G/\sqrt{T}$ in (7) vanishes, if $w_{\mathcal{S}}^*$ be a minimizer of the unconstrained problem, i.e., $\nabla\hat{F}(w_{\mathcal{S}}^*, \mathcal{S}) = 0$.

### 3.2 Generalization error

We derive our generalization bounds for MAML by establishing its algorithmic stability properties. The stability approach has been used widely to characterize the generalization properties for optimization algorithms such as stochastic gradient descent [18] or differentially private methods [33]. These arguments are based on showing the uniform stability of algorithms [17] which we restate it here.

**Definition 1** ([17]). *Consider the problem of minimizing the empirical function $\hat{\mathcal{L}}(w, \mathcal{H})$ for some dataset $\mathcal{H}$. A randomized algorithm $\mathcal{A}$ with output $w_{\mathcal{H}}$ given dataset $\mathcal{H}$ is called $\gamma$-uniformly stable if the following condition holds: Take the dataset $\tilde{\mathcal{H}}$ which is the same as $\mathcal{H}$, except at one data points. Then, we have $\sup_{\tilde{z} \in \mathcal{Z}} \mathbb{E}_{\mathcal{A}}[|\ell(w_{\mathcal{H}}, \tilde{z}) - \ell(w_{\tilde{\mathcal{H}}}, \tilde{z})|] \leq \gamma$, where the expectation is taken over the randomness of $\mathcal{A}$.*

The above definition captures the stability of an algorithm. Specifically, it states that Algorithm $\mathcal{A}$ is $\gamma$-stable, if the resulting loss of its outputs, when it is run using to two different datasets that only differ in one data point, are at most $\gamma$ away from each other. Note that the above definition holds if the difference between the losses *evaluated at any point $\tilde{z}$* is bounded by $\gamma$. The main importance of this definition is its connection with generalization error. In particular, it can be shown that if an algorithm is $\gamma$-uniformly stable and "symmetric", then its generalization error is bounded above by $\gamma$; see, e.g., [17]. Next, we formally state the definition of a symmetric algorithm.

**Definition 2.** *An algorithm $\mathcal{A} : \mathcal{Z}^n \to \mathbb{R}^d$ is called symmetric, if for any $\mathcal{S} \subset \mathcal{Z}^n$, the distribution of its output, i.e., $\mathcal{A}(\mathcal{S})$, does not depend on the ordering of elements of $\mathcal{S}$, i.e., if we take $\mathcal{S}'$ as a permutation of $\mathcal{S}$, the distribution of $\mathcal{A}(\mathcal{S})$ and $\mathcal{A}(\mathcal{S}')$ would be similar.*

Note that Definition 1 is useful for the case where we measure the performance of a model $w$ by its loss function over a sample, i.e., $\ell(w, \tilde{z})$. However, in this paper we measure the performance of a model by looking at its loss after one step of SGD which involves $K$ data points, as defined in (6). Therefore, we cannot directly use Definition 1 for characterizing the generalization error of MAML. In fact, in what follows, we first propose a modified version of the uniform stability definition, which is compatible with our setting, and then show how such stability could lead to generalization bounds for MAML-type algorithms.

**Definition 3.** *Consider the problem in* (4)*. A randomized algorithm $\mathcal{A}$ with output $w_{\mathcal{S}}$ given dataset $\mathcal{S}$ is called $(\gamma, K)$-uniformly stable if the following condition holds for any $i \in \{1, \ldots m\}$: Take the dataset $\tilde{\mathcal{S}}$ which is the same as $\mathcal{S}$, except that $\tilde{\mathcal{S}}_i^{in}$ and $\tilde{\mathcal{S}}_i^{out}$ differ from $\mathcal{S}_i^{in}$ and $\mathcal{S}_i^{out}$ in at most $K$ and one data points, respectively. Then, for any $\tilde{z} \in \mathcal{Z}$ and any $K$ distinct points $\{z_1, ..., z_K\}$ in $\mathcal{Z}$,*

$$\mathbb{E}_{\mathcal{A}} \left[ \left| \ell \left( w_{\mathcal{S}} - \alpha \nabla \hat{\mathcal{L}}(w_{\mathcal{S}}, \{z_j\}_{j=1}^K), \tilde{z} \right) - \ell \left( w_{\tilde{\mathcal{S}}} - \alpha \nabla \hat{\mathcal{L}}(w_{\tilde{\mathcal{S}}}, \{z_j\}_{j=1}^K), \tilde{z} \right) \right| \right] \leq \gamma,$$

*where the expectation is taken over the randomness of $\mathcal{A}$.*

A few remarks about the above definition follow. First, one might wonder, why it is needed to change $K$ points of the set $\mathcal{S}_i^{in}$, while we change only one point of the set $\mathcal{S}_i^{out}$. Note that, going from (1) to (3), the expectation $\mathbb{E}_{D_i^{test}}[.]$ is replaced by the sum over all $\binom{n}{K}$ possible batches $\mathcal{D}_i^{in}$ of size $K$ from $\mathcal{S}_i^{in}$. In other words, for the empirical sum in (3), each batch $\mathcal{D}_i^{in}$ can be seen as a data unit. That said, and similar to Definition 1, to characterize the stability, we need to change one data unit which is one batch of size $K$. That is why we change $K$ data points of $\mathcal{S}_i^{in}$ in the definition of $(\gamma, K)$-uniformly stability. On the other hand, we replace $\mathcal{L}_i(.) = \mathbb{E}_{z \sim p_i}[\ell(., z)]$ in (1) with a sum over $n$ points of $\mathcal{S}_i^{out}$ in (3), and thus, for this one, each data unit is just a single data point. So, similar to Definition 1, we just change one data point for the set $\mathcal{S}_i^{out}$.

Second, it is worth comparing this definition with the other definition given for stability of meta-learning algorithms in [28]. In that paper, the definition of stability is based on modifying the *whole dataset $\mathcal{S}_i$* rather than what we do here which is changing just $K + 1$ points. While taking such a definition makes the analysis relatively simpler, it prohibits us from characterizing the dependence of generalization error on $n$, and hence the resulting upper bound for generalization error would be larger. We will come back to this point later when we derive the stability of MAML with respect to Definition 3 and compare it with the one obtained in [28].

As we discussed, the main reason that we are interested in the uniform stability of an algorithm is its connection with generalization error. In the next theorem, we formalize this connection for MAML formulation and show that if an Algorithm $\mathcal{A}$ is $(\gamma, K)$-uniformly stable and symmetric, then its output generalization error is bounded above by $\gamma$. The proof of this result is available in Appendix C.

**Theorem 1.** *Consider the population and empirical losses defined in* (2) *and* (4)*, respectively. If Assumption 2 holds and $\mathcal{A}$ is a (possibly randomized) symmetric and $(\gamma, K)$-uniformly stable algorithm with output $w_{\mathcal{S}} \in \mathcal{W}$, then $\mathbb{E}_{\mathcal{A}, \mathcal{S}} \left[ F(w_{\mathcal{S}}) - \hat{F}(w_{\mathcal{S}}, \mathcal{S}) \right] \leq \gamma$.*

This result shows that if we prove a symmetric algorithm is $(\gamma, K)$-uniformly stable as defined in Definition 3, then we can bound its output model generalization error by $\gamma$. Hence, to characterize the generalization error of the model trained by MAML algorithm, we only need to capture the uniform stability parameter of MAML. Before stating this result, it is worth noting that while we limit our focus to MAML in this paper, Definition 3 and Theorem 1 could provide a framework for studying the generalization properties of a broader class of gradient-based meta-learning algorithms such as Reptile [34], First-order MAML [1], and Hessian-Free MAML [13].

**Theorem 2.** *If Assumption 1 holds, then MAML (Algorithm 1) with both last iterate and average iterate outputs and with $\alpha \leq \min\{\frac{1}{2L}, \frac{\mu}{8\rho G}\}$ and $\beta_t \leq \frac{1}{4L + 2\alpha\rho G}$ is $(\gamma, K)$-uniformly stable, where $\gamma := \mathcal{O}(1) \frac{G^2 (1 + \alpha L K)}{mn\mu}$.*

According to the above discussion, the result of Theorem 2 guarantees that the generalization error of MAML solution decays by a factor of $\mathcal{O}(K/mn)$, where $m$ is the number of tasks in the training set

and $n$ is the number of available samples per task. The classic lower bound for SGD over strongly convex functions translates to a $\mathcal{O}(1/mn)$ lower bound in our setting. Hence, our bound is tight in the small $K$ regime, which is generally the case in few-shot learning problems. However, one shortcoming of this result is that it is not tight in the large $K$ regime. In Appendix E we show how we could improve this result for the large $K$ regime. However, throughout the paper, we keep our discussion limited to the small $K$ regime.

**Remark 1.** *If instead of using our uniform stability definition (i.e., Definition 3), one uses the stability definition given in [28], the resulted stability constant $\gamma$ would be proportional to $(1/m)$ rather than $(1/mn)$. In fact, our proposed uniform-stability definition empowers us to obtain a better bound and indicates the role of number of samples per task $n$ in the generalization error.*

**Remark 2.** *The algorithmic stability technique is mainly limited to the convex setting, since, in the nonconvex case, we need to keep learning rate very small to obtain meaningful generalization results which makes it impractical (Check Appendix G for further discussions on this matter). In fact, the main reason that we assume $\ell$ is strongly convex and $\alpha \leq \Omega(\mu)$ is to ensure that the meta-objective is convex, as, in general, relaxing any of these two could lead to a nonconvex meta-objective function. However, these two assumptions together make the objective function strongly convex, which is not necessarily needed in our analysis. In fact, if we assume that $\ell$ and the meta-function are convex (but not necessarily strongly convex), we could still use Definition 3 to derive similar generalization bounds.*

Putting Proposition 1 and Theorem 2 together, we obtain the following result on the excess population loss of MAML algorithm. We only report the result for the averaged iterates here, but one can obtain the result for the last iterate similarly by using Proposition 1.

**Proposition 2.** *Consider the function $F$ defined in (2) with $\alpha \leq \min\{\frac{1}{2L}, \frac{\mu}{8\rho G}\}$. If Assumptions 1 and 2 hold, then the average of iterates generated by MAML (Algorithm 1) with $\beta_t = \min(\frac{1}{4L+2\alpha\rho G}, \frac{8}{\mu(t+1)})$ after $T$ iterations satisfies*

$$\mathbb{E}_{\mathcal{A},\mathcal{S}} \left[ F(\bar{w}^T) - \min_{\mathcal{W}} F \right] \leq \mathcal{O}(1)\frac{G^2}{\mu} \left( \frac{\log(T) + L/\mu}{T} + \frac{1 + \alpha LK}{mn} \right),$$

*where the expectation is taken over the sampling of $\mathcal{S}$ and the randomness of MAML algorithm.*

As an immediate application, the following corollary characterizes MAML test error.

**Corollary 1.** *Under the premise of Proposition 2, MAML algorithm after $T = \tilde{\mathcal{O}}(mnL/\mu)$ iterations returns $\bar{w}^T$ such that $\mathbb{E}_{\mathcal{A},\mathcal{S}} \left[ F(\bar{w}^T) - \min_{\mathcal{W}} F \right] \leq \mathcal{O} \left( G^2(1 + \alpha LK)/(mn\mu) \right).$*

### 3.3 Generalization to an unseen task

As we discussed in Section 2, another generalization measure is how the model trained with respect to the empirical problem in (4) performs on a new and unseen task $\mathcal{T}_{m+1}$ with corresponding distribution $p_{m+1}$. To state our result for this case, we first need to introduce the following distance notion between probability distributions.

**Definition 4.** *For two distributions $P$ and $Q$, defined over the sample space $\Omega$ and $\sigma$-field $\mathcal{F}$, the total variation distance is defined as $\|P - Q\|_{TV} := \sup_{A \in \mathcal{F}} |P(A) - Q(A)|$.*

It is well-known that the total variation distance admits the following characterization

$$\|P - Q\|_{TV} = \sup_{f:0 \leq f \leq 1} \mathbb{E}_{x \sim P}[f(x)] - \mathbb{E}_{x \sim Q}[f(x)]. \tag{8}$$

Also, we require the following boundedness assumption for our result.

**Assumption 3.** *For any $z \in \mathcal{Z}$, the function $\ell(., z)$ is $M$-bounded over $\mathcal{W}$.*

Considering these assumptions, we are ready to state our result for the case when the task at test time is a new task and is not observed during training.

**Theorem 3.** *Consider the population losses defined in (1) and (2). Suppose Assumptions 1, 2 and 3 hold. Then, for any $w \in \mathcal{W}$, we have*

$$|F_{m+1}(w) - F(w)| \leq D(p_{m+1}, \{p_i\}_{i=1}^m), \tag{9}$$

*where*

$$D(p_{m+1}, \{p_i\}_{i=1}^m) := \frac{4\alpha G^2}{m} \sum_{i=1}^m \|p_{m+1} - p_i\|_{TV} + (M + 2\alpha G^2)\|p_{m+1} - \frac{1}{m}\sum_{i=1}^m p_i\|_{TV}. \quad (10)$$

While the proof is provided in detail in Appendix F, here we discuss a sketch of it to highlight the main technical contributions. To simplify the notation here, let us assume $m = 1$, meaning that $p_1$ is the distribution used for training and $p_2$ is the distribution corresponding to the new task. Note that we aim to bound $|F_2(w) - F_1(w)|$. Recalling the definition of population loss (2), we need to bound the following expression (we drop the absolute value due to symmetry)

$$\mathbb{E}_{\{z_j^2 \sim p_2\}_{j=1}^K, \tilde{z}^2 \sim p_2}[l(w - \alpha\nabla\hat{\mathcal{L}}(w, \{z_j^2\}_j), \tilde{z}^2)] - \mathbb{E}_{\{z_j^1 \sim p_1\}_{j=1}^K, \tilde{z}^1 \sim p_1}[l(w - \alpha\nabla\hat{\mathcal{L}}(w, \{z_j^1\}_j), \tilde{z}^1)]. \quad (11)$$

Notice that this difference can be cast as $\mathbb{E}_{(\{z_j\}_{j=1}^K, \tilde{z}) \sim p_2^{K+1}}[X] - \mathbb{E}_{(\{z_j\}_{j=1}^K, \tilde{z}) \sim p_1^{K+1}}[X]$, with $X := l\left(w - \alpha\nabla\hat{\mathcal{L}}(w, \{z_j\}_{j=1}^K), \tilde{z}\right)$. As a result, a naive approach would be using Lipschitz and boundedness properties of $l$ (Assumptions 1 and 3) along with (8) to obtain a bound depending on $\|p_1^{K+1} - p_2^{K+1}\|_{TV} = \mathcal{O}(K)\|p_1 - p_2\|_{TV}$. However, this bound is not tight as it grows with $K$.

To address this issue, we exploit a coupling technique. Note that the expression in (11) does not depend on the joint distribution of $z_j^1$ and $z_j^2$, and instead, it only depends on the marginal distribution of $z_j^1$ and $z_j^2$. That said, for each $j$, we assume that $z_j^1$ and $z_j^2$ are sampled from a distribution $\mu$ on $\mathcal{Z} \times \mathcal{Z}$ such that $z_j^1 \sim p_1$, $z_j^2 \sim p_2$, and $\mu(z_j^1 \neq z_j^2) = \|p_1 - p_2\|_{TV}$. Such a coupling exists and is called *maximal coupling* of $p_1$ and $p_2$ [35]. Using this idea, as we show in Appendix F, we can eliminate the dependence on $K$, and as a result, the upper bound in (9) is independent of number of available labeled samples at test time denoted by $K$.

**Remark 3.** *Note that the terms $\frac{1}{m}\sum_{i=1}^m \|p_{m+1} - p_i\|_{TV}$ and $\|p_{m+1} - \frac{1}{m}\sum_{i=1}^m p_i\|_{TV}$ in $D(p_{m+1}, \{p_i\}_{i=1}^m)$ come from the fact that we consider uniform distribution over tasks in the empirical problem (4). In particular, if we instead consider the empirical problem $\arg\min_{w \in \mathcal{W}} \sum_{i=1}^m q_i\hat{F}_i(w, \mathcal{S}_i)$, for some non-negative weights $q_i$ with $\sum_{i=1}^m q_i = 1$, then $D(p_{m+1}, \{p_i\}_{i=1}^m)$ on the right hand side of (9) would change to*

$$(M + 2\alpha G^2)\|p_{m+1} - \sum_{i=1}^m q_i p_i\|_{TV} + 12\alpha G^2 \sum_{i=1}^m q_i\|p_{m+1} - p_i\|_{TV}.$$

*This result shows that by changing the training problem we can achieve a lower generalization error for MAML, if we have some information about the distribution $p_{m+1}$ at training time. For instance, if we know $p_{m+1}$ will be much closer to $p_1$ compared to $p_2$, making the weight of $p_1$ larger than $p_2$ would decrease the generalization error of MAML.*

**Corollary 2.** *Recall the population loss $F_{m+1}$ defined in (2) and $D(p_{m+1}, \{p_i\}_{i=1}^m)$ defined in Theorem 3. Let $\mathcal{A}$ be an algorithm for solving the empirical problem (4) which achieves $\epsilon$ excess risk, i.e., $\mathbb{E}_{\mathcal{A}, \mathcal{S}}[F(\mathcal{A}(\mathcal{S}))] - \min_{\mathcal{W}} F \leq \epsilon$. If Assumptions 1, 2 and 3 hold, then algorithm $\mathcal{A}$ finds a model $w_{\mathcal{S}}$ which achieves $\epsilon + D(p_{m+1}, \{p_i\}_{i=1}^m)$ excess loss with respect to $F_{m+1}$,*

$$\mathbb{E}_{\mathcal{A}, \mathcal{S}}[F_{m+1}(w_{\mathcal{S}})] - \min_{\mathcal{W}} F_{m+1} \leq \epsilon + 2D(p_{m+1}, \{p_i\}_{i=1}^m).$$

This corollary and Proposition 2 together imply that the MAML algorithm's test error with respect to the new task $\mathcal{T}_{m+1}$ is $\mathcal{O}(1)\left(\frac{1}{mn} + D(p_{m+1}, \{p_i\}_{i=1}^m)\right)$. As a result, if the new task's distribution $p_{m+1}$ is sufficiently close to the other tasks' distributions, MAML will have a low test error on the new unseen task. On the other hand, if $p_{m+1}$ is far from $p_1, \ldots, p_m$ in TV distance, then test error of the model trained $\{\mathcal{T}_{i=1}^m\}$ over $\mathcal{T}_{m+1}$ could be potentially large. In Appendix F.2 we show how this result can be extended to the case that the task at test time is generated from a distribution over both recurring tasks $\{\mathcal{T}_i\}_{i=1}^m$ and the unseen task $\mathcal{T}_{m+1}$.

## 4 Conclusion and future work

In this work, we studied the generalization of MAML algorithm in two key cases: $a)$ when the test time task is a recurring task from the ones observed during the training stage, $b)$ when it is a

new and unseen one. For the first one, and under strong convexity assumption, we showed that the generalization error improves as the number of tasks or the number of samples per task increases. For the second case, we showed that when the distance between the unseen task's distribution and the distributions of training tasks is sufficiently small, the MAML output generalizes well to the new task revealed at test time.

While we focused on the convex case in this paper, deriving generalization bounds when the meta-function is nonconvex is a natural future direction to explore. However, this could be challenging since the generalization of gradient methods is not well understood in the nonconvex setting even for the classic supervised learning problem.

## 5    Acknowledgment

Alireza Fallah acknowledges support from the Apple Scholars in AI/ML PhD fellowship and the MathWorks Engineering Fellowship. This research is sponsored by the United States Air Force Research Laboratory and the United States Air Force Artificial Intelligence Accelerator and was accomplished under Cooperative Agreement Number FA8750-19-2-1000. The views and conclusions contained in this document are those of the authors and should not be interpreted as representing the official policies, either expressed or implied, of the United States Air Force or the U.S. Government. The U.S. Government is authorized to reproduce and distribute reprints for Government purposes notwithstanding any copyright notation herein. This research of Aryan Mokhtari is supported in part by NSF Grant 2007668, ARO Grant W911NF2110226, the Machine Learning Laboratory at UT Austin, and the NSF AI Institute for Foundations of Machine Learning.

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
