# Appendix

## A    Intermediate Results

In this section we list a number of results that will be helpful in proofs of our main results.

**Lemma 2** (From [36] with modifications). *Let $\phi$ be a $\gamma$-strongly convex and $\eta$-smooth function which its gradient is bounded by $\tilde{G}$ over the convex and closed set $\mathcal{W}$. Then, we have*

$$\frac{\lambda}{2}\|w - w^*\|^2 \le \phi(w) - \phi(w^*) \le \frac{L}{2}\|w - w^*\|^2 + \tilde{G}\|w - w^*\|. \tag{12}$$

*Proof.* Recalling the definition of strong convexity and smoothness, we have

$$\frac{\lambda}{2}\|w - w^*\|^2 + \nabla\phi(w^*)^\top(w - w^*) \le \phi(w) - \phi(w^*) \le \frac{L}{2}\|w - w^*\|^2 + \tilde{G}\|w - w^*\| + \nabla\phi(w^*)^\top(w - w^*). \tag{13}$$

Since $w^* = \arg\min_{\mathcal{W}} \phi$, we have $\nabla\phi(w^*)^\top(w - w^*) \ge 0$, and hence from the left hand side of (13), we immediately obtain the left hand side of (12). To obtain the right hand side, it just suffices to use the bounded gradient assumption along with Cauchy–Schwarz inequality:

$$\nabla\phi(w^*)^\top(w - w^*) \le \tilde{G}\|w - w^*\|.$$

$\square$

**Lemma 3.** *Suppose the conditions in Assumption 1 are satisfied. Then, with $\alpha \le 1/L$, and for any batch $\mathcal{D}$ and $z \in \mathcal{Z}$, we have*

$$\left\|\nabla\ell\left(w - \alpha\nabla\hat{\mathcal{L}}(w, \mathcal{D}), z\right)\right\| \le 2G. \tag{14}$$

*for any $w \in \mathcal{W}$. Furthermore, if we take $v \in \mathcal{W}$ as well, we have*

$$\left|\ell\left(w - \alpha\nabla\hat{\mathcal{L}}(w, \mathcal{D}), z\right) - \ell\left(v - \alpha\nabla\hat{\mathcal{L}}(v, \mathcal{D}), z\right)\right| \le 4G\|w - v\|. \tag{15}$$

*Proof.* First, note that

$$\left\|\nabla\ell\left(w - \alpha\nabla\hat{\mathcal{L}}(w, \mathcal{D}), z\right)\right\| \le \|\nabla\ell(w, z)\| + \alpha L\|\hat{\mathcal{L}}(w, \mathcal{D})\|$$
$$\le (1 + \alpha L)G \le 2G, \tag{16}$$

where the first inequality follows from smoothness of $\ell(., \tilde{z})$ for any $\tilde{z}$, and the second inequality is obtained using the bounded gradient assumption. To show (15), let us define $\psi(w) = \ell\left(w - \alpha\nabla\hat{\mathcal{L}}(w, \mathcal{D}), z\right)$ for any $w \in \mathcal{W}$. Note that

$$\psi(w) - \psi(v) = \int_0^1 \nabla\psi(v + s(w - v))^\top(w - v)ds, \tag{17}$$

and hence,

$$|\psi(w) - \psi(v)| \le \int_0^1 \|\nabla\psi(v + s(w - v))\| \cdot \|w - v\|ds$$

$$= \|w - v\| \int_0^1 \left\|\left(I - \alpha\nabla^2\hat{\mathcal{L}}(v + s(w - v), \mathcal{D})\right)\nabla\ell\left(v + s(w - v) - \alpha\nabla\hat{\mathcal{L}}(v + s(w - v), \mathcal{D}), z\right)\right\|ds$$

$$\le 2\|w - v\| \int_0^1 \left\|\nabla\ell\left(v + s(w - v) - \alpha\nabla\hat{\mathcal{L}}(v + s(w - v), \mathcal{D}), z\right)\right\|ds, \tag{18}$$

where the last inequality follows from $\|\nabla^2\hat{\mathcal{L}}(v + s(w - v), \mathcal{D})\| \le L$ and $\alpha \le 1/L$. Therefore, it suffices to bound

$$\left\|\nabla\ell\left(v + s(w - v) - \alpha\nabla\hat{\mathcal{L}}(v + s(w - v), \mathcal{D}), z\right)\right\|.$$

Using the fact that $\mathcal{W}$ is convex, we have $v + s(w - v) \in \mathcal{W}$, and hence we could use the same approach in (16) and complete the proof. $\square$

**Lemma 4.** *Suppose Assumptions 1 and 3 hold. Then, with $\alpha \leq 1/L$, and for any batch $\mathcal{D}$ and $z \in \mathcal{Z}$, we have*

$$\left\| \ell \left( w - \alpha \nabla \hat{\mathcal{L}}(w, \mathcal{D}), z \right) \right\| \leq M + 2\alpha G^2, \tag{19}$$

*for any $w \in \mathcal{W}$.*

*Proof.* Let $h(\eta) := \ell \left( w - \eta \nabla \hat{\mathcal{L}}(w, \mathcal{D}), z \right)$. Using Lemma 3, it is easy to verify that $|h'(\eta)| \leq 2G^2$, and hence, using Mean-value Theorem, we have $|h(\alpha) - h(0)| \leq 2\alpha G^2$. This result, along with the fact that $|h(0)| = |\ell(w, z)| \leq M$ by Assumption 3 completes the proof. $\qquad \square$

As we stated in Section 2, MAML uses an unbiased gradient estimate at each iteration. The next lemma provides an upper bound on the variance of such estimate.

**Lemma 5.** *Consider the function $\hat{F}_i(., \mathcal{S}_i)$ defined in (4) with $\alpha \leq \frac{1}{L}$. Suppose the conditions in Assumption 1 are satisfied. Recall that for batches $\mathcal{D}_i^{in} \subset \mathcal{S}_i^{in}$ with size $K$ and $\mathcal{D}_i^{out} \subset \mathcal{S}_i^{out}$ with size $b$,*

$$g_i(w; \mathcal{D}_i^{in}, \mathcal{D}_i^{out}) = \left( I_d - \alpha \nabla^2 \hat{\mathcal{L}}(w, \mathcal{D}_i^{in}) \right) \nabla \hat{\mathcal{L}} \left( w - \alpha \nabla \hat{\mathcal{L}}(w, \mathcal{D}_i^{in}), \mathcal{D}_i^{out} \right)$$

*is an unbiased estimate of $\nabla \hat{F}_i(w, \mathcal{S}_i)$. Then, for any $w \in \mathcal{W}$, we have*

$$\|g_i(w; \mathcal{D}_i^{in}, \mathcal{D}_i^{out})\| \leq 4G,$$

$$\mathbb{E}_{\mathcal{D}_i^{in}, \mathcal{D}_i^{out}} \left[ \left\| g_i(w; \mathcal{D}_i^{in}, \mathcal{D}_i^{out}) - \nabla \hat{F}_i(w, \mathcal{S}_i) \right\|^2 \right] \mathcal{O}(1) G^2 \left( \frac{\alpha^2 L^2}{K} + \frac{1}{b} \right).$$

*Proof.* Recall from Lemma 3 that

$$\left\| \nabla \ell \left( w - \alpha \nabla \hat{\mathcal{L}}(w, \mathcal{D}_i^{in}), z \right) \right\| \leq 2G \tag{21}$$

As a result, we have

$$\|g_i(w; \mathcal{D}_i^{in}, \mathcal{D}_i^{out})\| \leq \|I_d - \alpha \nabla^2 \hat{\mathcal{L}}(w, \mathcal{D}_i^{in})\| \cdot \|\nabla \hat{\mathcal{L}} \left( w - \alpha \nabla \hat{\mathcal{L}}(w, \mathcal{D}_i^{in}), \mathcal{D}_i^{out} \right) \|$$
$$\leq (1 + \alpha L) 2G \leq 4G.$$

To show the second result, we first claim

$$\mathbb{E}_{\mathcal{D}_i^{in}} \left[ \left\| g_i(w; \mathcal{D}_i^{in}, \mathcal{S}_i^{out}) - g_i(w; \mathcal{S}_i^{in}, \mathcal{S}_i^{out}) \right\|^2 \right] \leq 36 \frac{\alpha^2 L^2 G^2}{K}. \tag{22}$$

To show this, let us define

$$e_H := \left( I_d - \alpha \nabla^2 \hat{\mathcal{L}}(w, \mathcal{D}_i^{in}) \right) - \left( I_d - \alpha \nabla^2 \hat{\mathcal{L}}(w, \mathcal{S}_i^{in}) \right) = \alpha \left( \nabla^2 \hat{\mathcal{L}}(w, \mathcal{S}_i^{in}) - \nabla^2 \hat{\mathcal{L}}(w, \mathcal{D}_i^{in}) \right)$$

$$e_G := \nabla \hat{\mathcal{L}} \left( w - \alpha \nabla \hat{\mathcal{L}}(w, \mathcal{D}_i^{in}), \mathcal{S}_i^{out} \right) - \nabla \hat{\mathcal{L}} \left( w - \alpha \nabla \hat{\mathcal{L}}(w, \mathcal{S}_i^{in}), \mathcal{S}_i^{out} \right).$$

Note that, by Assumption 1, we have

$$\|e_H\| \leq 2\alpha L, \quad \|e_G\| \leq \alpha L \|\nabla \hat{\mathcal{L}}(w, \mathcal{D}_i^{in}) - \nabla \hat{\mathcal{L}}(w, \mathcal{S}_i^{in})\| \leq 2\alpha L G. \tag{24}$$

In addition, using the fact that batch $\mathcal{D}_i^{in}$ is chosen uniformly at random, we have

$$\mathbb{E}_{\mathcal{D}_i^{in}}[\|e_H\|^2] \leq \alpha^2 \frac{L^2}{K} \cdot \frac{n - K}{n - 1}, \quad \mathbb{E}_{\mathcal{D}_i^{in}}[\|e_G\|^2] \leq \alpha^2 L^2 \frac{G^2}{K} \cdot \frac{n - K}{n - 1}. \tag{25}$$

Next, note that

$$g_i(w; \mathcal{D}_i^{in}, \mathcal{S}_i^{out}) - g_i(w; \mathcal{S}_i^{in}, \mathcal{S}_i^{out})$$
$$= e_H \nabla \hat{\mathcal{L}} \left( w - \alpha \nabla \hat{\mathcal{L}}(w, \mathcal{S}_i^{in}), \mathcal{S}_i^{out} \right) + e_G \left( I_d - \alpha \nabla^2 \hat{\mathcal{L}}(w, \mathcal{S}_i^{in}) \right) + e_G e_H.$$

Hence, using Cauchy-Schwarz inequality along with (21), we have

$$
\mathbb{E}_{\mathcal{D}_i^{\text{in}}}\left[\left\|g_i(w;\mathcal{D}_i^{\text{in}},\mathcal{S}_i^{\text{out}})-g_i(w;\mathcal{S}_i^{\text{in}},\mathcal{S}_i^{\text{out}})\right\|^2\right]
$$
$$
\leq 3(2G)^2\mathbb{E}_{\mathcal{D}_i^{\text{in}}}[\|e_H\|^2]+3(1+\alpha L)^2\mathbb{E}_{\mathcal{D}_i^{\text{in}}}[\|e_G\|^2]+3\mathbb{E}_{\mathcal{D}_i^{\text{in}}}[\|e_G e_H\|^2]
$$
$$
\leq 12G^2\mathbb{E}_{\mathcal{D}_i^{\text{in}}}[\|e_H\|^2]+12\mathbb{E}_{\mathcal{D}_i^{\text{in}}}[\|e_G\|^2]+12\alpha^2 L^2 G^2\mathbb{E}_{\mathcal{D}_i^{\text{in}}}[\|e_H\|^2].
$$

where the last inequality is obtained using (24) and $\alpha L \leq 1$. Now, using (24), we have

$$
\mathbb{E}_{\mathcal{D}_i^{\text{in}}}\left[\left\|g_i(w;\mathcal{D}_i^{\text{in}},\mathcal{S}_i^{\text{out}})-g_i(w;\mathcal{S}_i^{\text{in}},\mathcal{S}_i^{\text{out}})\right\|^2\right]
$$
$$
\leq 12(2+\alpha^2 L^2)\frac{n-K}{n-1}\cdot\frac{\alpha^2 L^2 G^2}{K}\leq 36\frac{\alpha^2 L^2 G^2}{K}.
$$

which is the desired claim. Using this result and (21), we imply

$$
\mathbb{E}_{\mathcal{D}_i^{\text{in}},\mathcal{D}_i^{\text{out}}}\left[\left\|g_i(w;\mathcal{D}_i^{\text{in}},\mathcal{D}_i^{\text{out}})-\nabla\hat{F}_i(w,\mathcal{S}_i)\right\|^2\right]
$$
$$
\leq \mathbb{E}_{\mathcal{D}_i^{\text{in}}}\left[\left\|g_i(w;\mathcal{D}_i^{\text{in}},\mathcal{S}_i^{\text{out}})-\nabla\hat{F}_i(w,\mathcal{S}_i)\right\|^2\right]+\frac{4G^2}{b}
$$
$$
\leq 4\left(36\frac{\alpha^2 L^2 G^2}{K}+\frac{G^2}{b}\right) \tag{26}
$$

and the proof is complete. $\qquad\square$

## B  Proof of Proposition 1

Recall that

$$
w^{t+1}=\prod_{\mathcal{W}}\left(w^t-\beta_t g^t\right),
$$

where $g^t:=\frac{1}{r}\sum_{i\in\mathcal{B}_t}g_i(w^t;\mathcal{D}_i^{t,\text{in}},\mathcal{D}_i^{t,\text{out}})$ is an unbiased estimate of $\hat{F}(w^t)$. Furthermore, by Lemma 5, we know that $\|g^t\|\leq\tilde{G}:=4G$. Also, recall from Lemma 1 that $\hat{F}$ is $\lambda$-strongly convex with $\lambda:=\mu/8$.

Let $\mathcal{F}^t$ be the $\sigma$-field generated by the information up to time $t$ (and not including iteration $t$, such as the randomness in $\mathcal{B}_t$, etc.) It is worth noting that $\mathbb{E}[g^t\mid\mathcal{F}^t]=\nabla\hat{F}(w^t)$.

First, we claim that similar to the proof of Lemma 1 in [30], we could show

$$
\mathbb{E}[\|w^{t+1}-w^*\|^2]\leq(1-2\beta_t\lambda)\mathbb{E}[\|w^t-w^*\|^2]+\beta_t^2\tilde{G}^2, \tag{27}
$$

where $w^*$ is the minimizer of $\hat{F}(.,\mathcal{S})$ over $\mathcal{W}$. To see this, and for the sake of completeness, let us recall the steps of the proof. Note that

$$
\mathbb{E}\left[\|w^{t+1}-w^*\|^2\right]=\mathbb{E}\left[\left\|\prod_{\mathcal{W}}\left(w^t-\beta_t g^t\right)-w^*\right\|^2\right]
$$
$$
\leq\mathbb{E}\left[\|w^t-\beta_t g^t-w^*\|^2\right] \tag{28}
$$
$$
=\mathbb{E}\left[\|w^t-w^*\|^2\right]-2\beta_t\mathbb{E}\left[\langle g^t,w^t-w^*\rangle\right]+\beta_t^2\mathbb{E}\left[\|g^t\|^2\right]
$$
$$
=\mathbb{E}\left[\|w^t-w^*\|^2\right]-2\beta_t\mathbb{E}\left[\left\langle\hat{F}(w^t),w^t-w^*\right\rangle\right]+\beta_t^2\mathbb{E}\left[\|g^t\|^2\right], \tag{29}
$$

where (28) follows from non-expansivity of projection and (29) comes from the fact that $w^t\in\mathcal{F}^t$ and $\mathbb{E}[g^t\mid\mathcal{F}^t]=\nabla\hat{F}(w^t)$. Now, having (29), and using $\|g^t\|\leq\tilde{G}$ along with the strong convexity

of $\hat{F}$, we have

$$\mathbb{E}\left[\left\|w^{t+1}-w^*\right\|^2\right] \leq \mathbb{E}\left[\left\|w^t-w^*\right\|^2\right] - 2\beta_t\mathbb{E}\left[\hat{F}\left(w^t\right) - \hat{F}\left(w^*\right) + \frac{\lambda}{2}\left\|w^t-w^*\right\|^2\right] + \beta_t^2\tilde{G}^2$$

$$\leq \mathbb{E}\left[\left\|w^t-w^*\right\|^2\right] - 2\beta_t\mathbb{E}\left[\frac{\lambda}{2}\left\|w^t-w^*\right\|^2 + \frac{\lambda}{2}\left\|w^t-w^*\right\|^2\right] + \beta_t^2\tilde{G}^2 \tag{30}$$

$$= (1 - 2\beta_t\lambda)\,\mathbb{E}\left[\left\|w^t-w^*\right\|^2\right] + \beta_t^2\tilde{G}^2,$$

where (30) follows from Lemma 2. Next, note that $\beta_t$ is given by

$$\beta_t = \left\{ \begin{array}{ll} \beta, & \text{for } t \leq t^* - 1 \\ \frac{1}{\lambda(t+1)}, & \text{for } t > t^* - 1 \end{array} \right., \quad \text{with } t^* := \lfloor\frac{1}{\beta\lambda}\rfloor.$$

For any $t \leq t^*$, from (27) and Lemma 2 in [30], we obtain

$$\mathbb{E}[\|w^t-w^*\|^2] \leq \frac{\tilde{G}^2}{\lambda^2} + \beta^2\tilde{G}^2 t \leq \frac{\tilde{G}^2(t+3)}{\lambda^2(t+1)}. \tag{31}$$

Also, note that, for $t \geq t^*$, we have

$$\mathbb{E}[\|w^{t+1}-w^*\|^2] \leq (1 - \frac{2}{t+1})\mathbb{E}[\|w^t-w^*\|^2] + \frac{\tilde{G}^2}{\lambda^2(t+1)^2}. \tag{32}$$

Hence, by induction, it can be seen that for any t, we have

$$\mathbb{E}[\|w^t-w^*\|^2] \leq \frac{\tilde{G}^2(t^*+3)}{\lambda^2(t+1)}. \tag{33}$$

Using Lemma 2 gives us (7).

To obtain the bound on the time-average iterate, first, we could similarly, modify the result in [31] to obtain

$$2\mathbb{E}[\hat{F}(\bar{w}^T) - \hat{F}(w^*)] \leq \frac{1}{T}\left(\|w^0-w^*\|^2(\frac{1}{\beta_1} - \lambda) + \sum_{t=1}^{T-1}\mathbb{E}[\|w^t-w^*\|^2](\frac{1}{\beta_{t+1}} - \frac{1}{\beta_t} - \lambda) + \tilde{G}^2\sum_{t=1}^{T}\beta_t\right).$$

It can be easily verified that for $\beta_t = \min(\beta, \frac{8}{\mu(t+1)})$, the term $\frac{1}{\beta_{t+1}} - \frac{1}{\beta_t} - \lambda$ is always non-positive. Hence, we have

$$\mathbb{E}[\hat{F}(\bar{w}^T) - \hat{F}(w^*)] \leq \|w^0-w^*\|^2\frac{1/\beta - \lambda}{T+1} + \frac{2\tilde{G}^2}{T+1}\sum_{t=0}^{T}\beta_t$$

$$\leq \mathcal{O}(1)\frac{\tilde{G}^2}{\lambda T}(1 + \log(T) - \log(t^*)) \leq \mathcal{O}(1)\frac{\tilde{G}^2}{\lambda T}\left(\frac{1}{\beta\lambda} + \log(T)\right), \tag{34}$$

where the last inequality follows from the fact that $4\tilde{G}^2/\lambda^2 \geq \|w^0-w^*\|^2$ (see Lemma 2 in [30] for the proof.)

## C  Proof of Theorem 1

To show the claim, it just suffices to show that for any $i$, we have

$$\mathbb{E}_{\mathcal{A},\mathcal{S}}\left[F_i(w_\mathcal{S}) - \hat{F}_i(w_\mathcal{S}, \mathcal{S}_i)\right] \leq \gamma. \tag{35}$$

Consider

$$\mathcal{S}_i^{\text{in}} = \{z_1^{\text{in}}, ..., z_n^{\text{in}}\}, \quad \mathcal{S}_i^{\text{out}} = \{z_1^{\text{out}}, ..., z_n^{\text{out}}\}.$$

To see this, first note that

$$F_i(w_\mathcal{S}) = \mathbb{E}_{\{z_j\}_{j=1}^K, \tilde{z}}\left[\ell\left(w_\mathcal{S} - \alpha\nabla\hat{\mathcal{L}}(w_\mathcal{S}, \{z_j\}_{j=1}^K), \tilde{z}\right)\right],$$

where $\{z_j\}_{j=1}^{K}$ are $K$ distinct points sampled from $p_i$ and $\tilde{z}$ is also independently sampled from $p_i$. By Assumption 2, we could assume $\tilde{z}$ is different from $K$ other points. Note that we have

$$\mathbb{E}_{\mathcal{S}}[F_i(w_{\mathcal{S}})] = \mathbb{E}_{\mathcal{S}, \{z_j\}_{j=1}^{K}, \tilde{z}} \left[ \ell \left( w_{\mathcal{S}} - \alpha \nabla \hat{\mathcal{L}}(w_{\mathcal{S}}, \{z_j\}_{j=1}^{K}), \tilde{z} \right) \right]. \tag{36}$$

Next, note that, we can write $\hat{F}_i(w_{\mathcal{S}}, \mathcal{S}_i)$ as

$$\hat{F}_i(w_{\mathcal{S}}, \mathcal{S}_i) = \frac{1}{\binom{n}{K} |\mathcal{S}_i^{\text{out}}|} \sum_{\substack{\{\zeta_j\}_{j=1}^{K} \subset [n] \\ \tilde{\zeta} \in [n]}} \ell \left( w_{\mathcal{S}} - \alpha \nabla \hat{\mathcal{L}}(w_{\mathcal{S}}, \{z_{\zeta_j}^{\text{in}}\}_{j=1}^{K}), z_{\tilde{\zeta}}^{\text{out}} \right).$$

Thus, we have

$$\mathbb{E}_{\mathcal{A}, \mathcal{S}}[\hat{F}_i(w_{\mathcal{S}}, \mathcal{S}_i)] = \frac{1}{\binom{n}{K} |\mathcal{S}_i^{\text{out}}|} \sum_{\substack{\{\zeta_j\}_{j=1}^{K} \subset [n] \\ \tilde{\zeta} \in [n]}} \mathbb{E}_{\mathcal{A}, \mathcal{S}} \left[ \ell \left( w_{\mathcal{S}} - \alpha \nabla \hat{\mathcal{L}}(w_{\mathcal{S}}, \{z_{\zeta_j}^{\text{in}}\}_{j=1}^{K}), z_{\tilde{\zeta}}^{\text{out}} \right) \right].$$

Notice that, $\{\zeta_j\}_{j=1}^{K}$ are all different, and hence, due to the symmetry, all the expectations on the RHS are equal. Hence, for a fixed $\{\zeta_j\}_{j=1}^{K} \subset [n]$ and $\tilde{\zeta} \in [n]$, we have

$$\mathbb{E}_{\mathcal{A}, \mathcal{S}}[\hat{F}_i(w_{\mathcal{S}}, \mathcal{S}_i)] = \mathbb{E}_{\mathcal{A}, \mathcal{S}} \left[ \ell \left( w_{\mathcal{S}} - \alpha \nabla \hat{\mathcal{L}}(w_{\mathcal{S}}, \{z_{\zeta_j}^{\text{in}}\}_{j=1}^{K}), z_{\tilde{\zeta}}^{\text{out}} \right) \right]$$
$$= \mathbb{E}_{\mathcal{A}, \mathcal{S}, \{z_j\}_{j=1}^{K}, \tilde{z}} \left[ \ell \left( w_{\mathcal{S}} - \alpha \nabla \hat{\mathcal{L}}(w_{\mathcal{S}}, \{z_{\zeta_j}^{\text{in}}\}_{j=1}^{K}), z_{\tilde{\zeta}}^{\text{out}} \right) \right] \tag{37}$$

Next, define the dataset $\tilde{\mathcal{S}}$ by substituting $z_{\zeta_j}^{\text{in}}$ with $z_j$, for all $j$, and $z_{\tilde{\zeta}}^{\text{out}}$ with $\tilde{z}$. It is straightforward to see that

$$\mathbb{E}_{\mathcal{A}, \mathcal{S}, \{z_j\}_{j=1}^{K}, \tilde{z}} \left[ \ell \left( w_{\mathcal{S}} - \alpha \nabla \hat{\mathcal{L}}(w_{\mathcal{S}}, \{z_{\zeta_j}^{\text{in}}\}_{j=1}^{K}), z_{\tilde{\zeta}}^{\text{out}} \right) \right] = \mathbb{E}_{\mathcal{A}, \mathcal{S}, \{z_j\}_{j=1}^{K}, \tilde{z}} \left[ \ell \left( w_{\tilde{\mathcal{S}}} - \alpha \nabla \hat{\mathcal{L}}(w_{\tilde{\mathcal{S}}}, \{z_j\}_{j=1}^{K}), \tilde{z} \right) \right]$$

Therefore, using (37), we obtain

$$\mathbb{E}_{\mathcal{A}, \mathcal{S}}[\hat{F}_i(w_{\mathcal{S}}, \mathcal{S}_i)] = \mathbb{E}_{\mathcal{A}, \mathcal{S}, \{z_j\}_{j=1}^{K}, \tilde{z}} \left[ \ell \left( w_{\tilde{\mathcal{S}}} - \alpha \nabla \hat{\mathcal{L}}(w_{\tilde{\mathcal{S}}}, \{z_j\}_{j=1}^{K}), \tilde{z} \right) \right]. \tag{38}$$

Putting (36) and (38) together, we have

$$\mathbb{E}_{\mathcal{A}, \mathcal{S}} \left[ F_i(w_{\mathcal{S}}) - \hat{F}_i(w_{\mathcal{S}}, \mathcal{S}_i) \right]$$
$$\leq \mathbb{E}_{\mathcal{A}, \mathcal{S}, \{z_j\}_{j=1}^{K}, \tilde{z}} \left[ \left| \ell \left( w_{\mathcal{S}} - \alpha \nabla \hat{\mathcal{L}}(w_{\mathcal{S}}, \{z_j\}_{j=1}^{K}), \tilde{z} \right) - \ell \left( w_{\tilde{\mathcal{S}}} - \alpha \nabla \hat{\mathcal{L}}(w_{\tilde{\mathcal{S}}}, \{z_j\}_{j=1}^{K}), \tilde{z} \right) \right| \right]$$
$$= \mathbb{E}_{\mathcal{S}, \{z_j\}_{j=1}^{K}, \tilde{z}} \left[ \mathbb{E}_{\mathcal{A}} \left[ \left| \ell \left( w_{\mathcal{S}} - \alpha \nabla \hat{\mathcal{L}}(w_{\mathcal{S}}, \{z_j\}_{j=1}^{K}), \tilde{z} \right) - \ell \left( w_{\tilde{\mathcal{S}}} - \alpha \nabla \hat{\mathcal{L}}(w_{\tilde{\mathcal{S}}}, \{z_j\}_{j=1}^{K}), \tilde{z} \right) \right| \right] \right] \tag{39}$$

where the last equality follows from Tonelli' theorem. Finally, note that since $\mathcal{A}$ is $(\gamma, K)$-uniformly stable, we could bound the the inner integral by $\gamma$, i.e.,

$$\mathbb{E}_{\mathcal{A}} \left[ \left| \ell \left( w_{\mathcal{S}} - \alpha \nabla \hat{\mathcal{L}}(w_{\mathcal{S}}, \{z_j\}_{j=1}^{K}), \tilde{z} \right) - \ell \left( w_{\tilde{\mathcal{S}}} - \alpha \nabla \hat{\mathcal{L}}(w_{\tilde{\mathcal{S}}}, \{z_j\}_{j=1}^{K}), \tilde{z} \right) \right| \right] \leq \gamma,$$

and thus, we obtain the desired result (35).

## D   Proof of Theorem 2

The stability definition says there is one $i$ such that the two datasets $\mathcal{S}$ and $\tilde{\mathcal{S}}$ differ only in the the two following terms:

- $\tilde{\mathcal{S}}_i^{\text{in}}$ differs from $\mathcal{S}_i^{\text{in}}$ in at most $K$ points. We show those $K$ samples by $\{z_j\}_{j=1}^{K}$ and $\{\tilde{z}_j\}_{j=1}^{K}$, respectively.

- $\tilde{\mathcal{S}}_i^{\text{out}}$ differs from $\mathcal{S}_i^{\text{out}}$ in at most one point. We show those by $\zeta$ and $\tilde{\zeta}$, respectively.

Let's consider two parallel processes of generating iterates $\{w^t\}$ and $\{\tilde{w}^t\}$ by using datasets $\mathcal{S}$ and $\tilde{\mathcal{S}}$, respectively. We use the tilde superscript to refer to the second process throughout the proof. Also, we use $D_i^{t,\text{out}}$ and $D_i^{t,\text{in}}$ to refer to indices of samples in $\mathcal{D}_i^{t,\text{out}}$ and $\mathcal{D}_i^{t,\text{in}}$, respectively. Also, with a slight abuse of notation, by $\hat{\mathcal{L}}(w^t, D_i^{\text{in/out}})$ we mean $\hat{\mathcal{L}}(w^t, \mathcal{D}_i^{\text{in/out}})$.

Note that the randomness of algorithm comes from the randomness in drawing batches at each iteration. We do a coupling argument here. We could assume the two parallel processes of generating iterates $\{w^t\}$ and $\{\tilde{w}^t\}$ use the same random machine for sampling batches. In other words, $\mathcal{B}_t = \tilde{\mathcal{B}}_t$, $D_i^{t,\text{out}} = \tilde{D}_i^{t,\text{out}}$, and $D_i^{t,\text{in}} = \tilde{D}_i^{t,\text{in}}$

For one particular realization:

- Let $u_t$ be the number of times that the index corresponding to sample $\zeta$ (or $\tilde{\zeta}$) is chosen in $D_i^{t,\text{out}}$. Note that this number could be zero if $i \notin \mathcal{B}_t$, and it could be greater than one if $i \in \mathcal{B}_t$ since $D_i^{t,\text{out}}$ is chosen with replacement.

- Let $v_t$ be the number of indices corresponding to the samples $\{z_j\}_{j=1}^K$ (or $\{\tilde{z}_j\}_{j=1}^K$) that appears in $D_i^{t,\text{in}}$. Again, this number could be zero if $i \notin \mathcal{B}_t$. Also, note that we take $D_i^{t,\text{in}}$ as a batch of $K$ different samples from $\mathcal{S}_i^{\text{in}}$, and hence, each one of $j$ indices appears at most one time in $D_i^{t,\text{in}}$.

The rest of the proof has three steps:

1. First, recall the definition of $b$ and $r$ from Alghorithm 1. We claim
$$\mathbb{E}[u_t] = \frac{br}{nm}, \quad \mathbb{E}[v_t] = \frac{K^2 r}{nm}. \tag{40}$$
The first one is easy to see. Task $i$ is in $\mathcal{B}_t$ with probability $r/m$, and if that happens, then $u_t$ would have a binomial distribution with mean $b/n$. To see the second one, note that
$$\mathbb{P}(v_t = j) = \binom{K}{j}\binom{n-K}{K-j},$$
and therefore,
$$\mathbb{E}[v_t | i \in \mathcal{B}_t] = \frac{1}{\binom{n}{K}} \sum_{j=0}^K j \binom{K}{j}\binom{n-K}{K-j}.$$
Using the fact that $\binom{K}{j} = \frac{K}{j}\binom{K-1}{j-1}$, we obtain
$$\mathbb{E}[v_t | i \in \mathcal{B}_t] = \frac{K}{\binom{n}{K}} \sum_{j=0}^K \binom{K-1}{j-1}\binom{n-K}{K-j}$$
$$= \frac{K}{\binom{n}{K}} \sum_{j=0}^{K-1} \binom{K-1}{j}\binom{(n-1)-(K-1)}{(K-1)-j}. \tag{41}$$
However, note that $\binom{K-1}{j}\binom{(n-1)-(K-1)}{(K-1)-j}$ is exactly the probability of $v_t = j$ if $K \to K-1$ and $n \to n-1$. Hence, the sum $\sum_{j=0}^{K-1} \binom{K-1}{j}\binom{(n-1)-(K-1)}{(K-1)-j}$ is equal to $\binom{n-1}{K-1}$, and plugging this into (41) gives us the second part of the claim (40).

2. Second, we claim that under Assumption 1 we have
$$\mathbb{E}_\mathcal{A}[\|w^T - \tilde{w}^T\|] \le \frac{4G}{mn}(1 + \alpha LK)\frac{16(2L + \rho\alpha G) + \mu}{\mu(2L + \rho\alpha G)}. \tag{42}$$
Before showing its proof, note that since $L \ge \mu$, this could be simplified as
$$\mathbb{E}_\mathcal{A}[\|w^T - \tilde{w}^T\|] \le \mathcal{O}(1)\frac{G}{mn\mu}(1 + \alpha LK). \tag{43}$$
Now, let's show why this is true. To simplify the notation, let us define $\psi(w; \mathcal{D}, z) := \ell\left(w - \alpha\nabla\hat{\mathcal{L}}(w, \mathcal{D}), z\right)$. We start by revisiting the following lemma from [18]:

**Lemma 6.** *Let $\phi$ be a $\lambda$-strongly convex and $\eta$-smooth function. Then, for any $\beta \leq \frac{2}{\lambda+\eta}$, we have*

$$\|(u - \beta\nabla\phi(u)) - (v - \beta\nabla\phi(v))\| \leq (1 - \frac{\beta\lambda\eta}{\lambda+\eta})\|u - v\|,$$

*for any $u$ and $v$.*

Next, recall from Lemma 1 that for any batch $\mathcal{D}$ and any $z \in \mathcal{Z}$, $\psi(w; \mathcal{D}, z)$ is $4L + 2\alpha\rho G$ smooth and $\mu/8$ strongly convex. Hence, using the above lemma, for any $j \in \mathcal{B}_t$ that $j \neq i$, we have

$$\|w_j^{t+1} - \tilde{w}_j^{t+1}\| \leq \left(1 - \beta_t \frac{2\mu(2L + \rho\alpha G)}{16(2L + \rho\alpha G) + \mu}\right)\|w^t - \tilde{w}^t\|. \tag{44}$$

Next, let us assume $i \in \mathcal{B}_t$. In this case, we have

$$\|w_i^{t+1} - \tilde{w}_i^{t+1}\| \leq \frac{1}{b} \sum_{z \in \mathcal{D}_i^{t,\text{out}}} \left\|\left(w^t - \beta_t\nabla\psi(w^t; \mathcal{D}_i^{t,\text{in}}, z)\right) - \left(\tilde{w}^t - \beta_t\nabla\psi(\tilde{w}^t; \tilde{\mathcal{D}}_i^{t,\text{in}}, z)\right)\right\|. \tag{45}$$

$$+ \frac{1}{b}\beta_t \sum_{z \in \tilde{\mathcal{D}}_i^{t,\text{out}}/\mathcal{D}_i^{t,\text{out}}} \left\|\nabla\psi(\tilde{w}^t; \tilde{\mathcal{D}}_i^{t,\text{in}}, z) - \nabla\psi(w^t; \mathcal{D}_i^{t,\text{in}}, z)\right\|. \tag{46}$$

For (46), note that we know by Lemma 5 that $\|\nabla\psi(w; , \mathcal{D}, z)\| \leq 4G$, and hence, since $|\tilde{\mathcal{D}}_i^{t,\text{out}}/\mathcal{D}_i^{t,\text{out}}| = u_t$, we could bound the second term by $8\beta_t G u_t/b$. As a result, we have

$$\|w_i^{t+1} - \tilde{w}_i^{t+1}\| \leq 8\beta_t G \frac{u_t}{b}$$
$$+ \frac{1}{b} \sum_{z \in \mathcal{D}_i^{t,\text{out}}} \left\|\left(w^t - \beta_t\nabla\psi(w^t; \mathcal{D}_i^{t,\text{in}}, z)\right) - \left(\tilde{w}^t - \beta_t\nabla\psi(\tilde{w}^t; \tilde{\mathcal{D}}_i^{t,\text{in}}, z)\right)\right\|. \tag{47}$$

Note that

$$\left\|\left(w^t - \beta_t\nabla\psi(w^t; \mathcal{D}_i^{t,\text{in}}, z)\right) - \left(\tilde{w}^t - \beta_t\nabla\psi(\tilde{w}^t; \tilde{\mathcal{D}}_i^{t,\text{in}}, z)\right)\right\|$$
$$\leq \left\|\left(w^t - \beta_t\nabla\psi(w^t; \mathcal{D}_i^{t,\text{in}}, z)\right) - \left(\tilde{w}^t - \beta_t\nabla\psi(\tilde{w}^t; \mathcal{D}_i^{t,\text{in}}, z)\right)\right\|$$
$$+ \beta_t \left\|\nabla\psi(\tilde{w}^t; \mathcal{D}_i^{t,\text{in}}, z) - \nabla\psi(\tilde{w}^t; \tilde{\mathcal{D}}_i^{t,\text{in}}, z)\right\|. \tag{48}$$

Let us bound the two terms on the RHS of (48) separately. First, similar to how we derived 44, we could bound the first term by

$$\left\|\left(w^t - \beta_t\nabla\psi(w^t; \mathcal{D}_i^{t,\text{in}}, z)\right) - \left(\tilde{w}^t - \beta_t\nabla\psi(\tilde{w}^t; \mathcal{D}_i^{t,\text{in}}, z)\right)\right\|$$
$$\leq \left(1 - \beta_t \frac{2\mu(2L + \rho\alpha G)}{16(2L + \rho\alpha G) + \mu}\right)\|w^t - \tilde{w}^t\|. \tag{49}$$

To bound the second term on the RHS of (48), note that

$$\left\|\nabla\psi(\tilde{w}^t; \mathcal{D}_i^{t,\text{in}}, z) - \nabla\psi(\tilde{w}^t; \tilde{\mathcal{D}}_i^{t,\text{in}}, z)\right\|$$
$$= \left\|(I - \alpha\nabla^2\hat{\mathcal{L}}(\tilde{w}^t, \mathcal{D}_i^{t,\text{in}}))\nabla\ell\left(\tilde{w}^t - \alpha\nabla\hat{\mathcal{L}}(\tilde{w}^t, \mathcal{D}_i^{t,\text{in}}), z\right)\right.$$
$$\left. - (I - \alpha\nabla^2\hat{\mathcal{L}}(\tilde{w}^t, \tilde{\mathcal{D}}_i^{t,\text{in}}))\nabla\ell\left(\tilde{w}^t - \alpha\nabla\hat{\mathcal{L}}(\tilde{w}^t, \tilde{\mathcal{D}}_i^{t,\text{in}}), z\right)\right\|$$
$$\leq \left\|\nabla\ell\left(\tilde{w}^t - \alpha\nabla\hat{\mathcal{L}}(\tilde{w}^t, \mathcal{D}_i^{t,\text{in}}), z\right) - \nabla\ell\left(\tilde{w}^t - \alpha\nabla\hat{\mathcal{L}}(\tilde{w}^t, \tilde{\mathcal{D}}_i^{t,\text{in}}), z\right)\right\| +$$
$$\alpha\left\|\nabla^2\hat{\mathcal{L}}(\tilde{w}^t, \mathcal{D}_i^{t,\text{in}})\nabla\ell\left(\tilde{w}^t - \alpha\nabla\hat{\mathcal{L}}(\tilde{w}^t, \mathcal{D}_i^{t,\text{in}}), z\right) - \nabla^2\hat{\mathcal{L}}(\tilde{w}^t, \tilde{\mathcal{D}}_i^{t,\text{in}})\nabla\ell\left(\tilde{w}^t - \alpha\nabla\hat{\mathcal{L}}(\tilde{w}^t, \tilde{\mathcal{D}}_i^{t,\text{in}}), z\right)\right\|$$
$$\leq (1 + \alpha L)\left\|\nabla\ell\left(\tilde{w}^t - \alpha\nabla\hat{\mathcal{L}}(\tilde{w}^t, \mathcal{D}_i^{t,\text{in}}), z\right) - \nabla\ell\left(\tilde{w}^t - \alpha\nabla\hat{\mathcal{L}}(\tilde{w}^t, \tilde{\mathcal{D}}_i^{t,\text{in}}), z\right)\right\| +$$
$$2\alpha G\left\|\nabla^2\hat{\mathcal{L}}(\tilde{w}^t, \mathcal{D}_i^{t,\text{in}}) - \nabla^2\hat{\mathcal{L}}(\tilde{w}^t, \tilde{\mathcal{D}}_i^{t,\text{in}})\right\|, \tag{50}$$

where, in the last inequality, we used Lemma 3 along with the third condition of Assumption 1. Hence, what remains is to bound the two terms in (50). To do so, notice that

$$\left\| \nabla \ell \left( \tilde{w}^t - \alpha \nabla \hat{\mathcal{L}}(\tilde{w}^t, \mathcal{D}_i^{t,\mathrm{in}}), z \right) - \nabla \ell \left( \tilde{w}^t - \alpha \nabla \hat{\mathcal{L}}(\tilde{w}^t, \tilde{\mathcal{D}}_i^{t,\mathrm{in}}), z \right) \right\|$$
$$\leq \alpha L \left\| \nabla \hat{\mathcal{L}}(\tilde{w}^t, \mathcal{D}_i^{t,\mathrm{in}}) - \nabla \hat{\mathcal{L}}(\tilde{w}^t, \tilde{\mathcal{D}}_i^{t,\mathrm{in}}) \right\| \leq 2\alpha L G \frac{v_t}{K}, \tag{51}$$

and

$$\left\| \nabla^2 \hat{\mathcal{L}}(\tilde{w}^t, \mathcal{D}_i^{t,\mathrm{in}}) - \nabla^2 \hat{\mathcal{L}}(\tilde{w}^t, \tilde{\mathcal{D}}_i^{t,\mathrm{in}}) \right\| \leq 2L \frac{v_t}{K}. \tag{52}$$

By plugging (51) and (52) into (50) and using $\alpha L \leq 1$, we have

$$\left\| \nabla \psi(\tilde{w}^t; \mathcal{D}_i^{t,\mathrm{in}}, z) - \nabla \psi(\tilde{w}^t; \tilde{\mathcal{D}}_i^{t,\mathrm{in}}, z) \right\| \leq 8\alpha L G \frac{v_t}{K}. \tag{53}$$

Substituting this bound and (49) into (48) and plugging the result into (47), we have

$$\| w_i^{t+1} - \tilde{w}_i^{t+1} \| \leq \left( 1 - \beta_t \frac{2\mu(2L + \rho\alpha G)}{16(2L + \rho\alpha G) + \mu} \right) \| w^t - \tilde{w}^t \| + 8\beta_t G(\frac{u_t}{b} + \alpha L \frac{v_t}{K}). \tag{54}$$

Using (54) and (44), we obtain

$$\| \frac{1}{r} \sum_{j \in \mathcal{B}_t} w_j^{t+1} - \frac{1}{r} \sum_{j \in \mathcal{B}_t} \tilde{w}_j^{t+1} \| \leq \left( 1 - \beta_t \frac{2\mu(2L + \rho\alpha G)}{16(2L + \rho\alpha G) + \mu} \right) \| w^t - \tilde{w}^t \| + 8\beta_t G(\frac{u_t}{rb} + \alpha L \frac{v_t}{rK}).$$

Since projections are non-expansive, we have

$$\| w^{t+1} - \tilde{w}^{t+1} \| \leq \left( 1 - \beta_t \frac{2\mu(2L + \rho\alpha G)}{16(2L + \rho\alpha G) + \mu} \right) \| w^t - \tilde{w}^t \| + 8\beta_t G(\frac{u_t}{rb} + \alpha L \frac{v_t}{rK}). \tag{55}$$

Taking an expectation from both sides and using (40), we get

$$\mathbb{E}_{\mathcal{A}}[\| w^{t+1} - \tilde{w}^{t+1} \|] \leq \left( 1 - \beta_t \frac{2\mu(2L + \rho\alpha G)}{16(2L + \rho\alpha G) + \mu} \right) \mathbb{E}_{\mathcal{A}}[\| w^t - \tilde{w}^t \|] + 8\frac{\beta_t G}{mn}(1 + \alpha L K). \tag{56}$$

Note that we can rewrite this bound as

$$\mathbb{E}_{\mathcal{A}}[\| w^{t+1} - \tilde{w}^{t+1} \|] \leq (1 - \beta_t \lambda) \mathbb{E}_{\mathcal{A}}[\| w^t - \tilde{w}^t \|] + \beta_t \eta,$$

where

$$\lambda := \frac{2\mu(2L + \rho\alpha G)}{16(2L + \rho\alpha G) + \mu}, \quad \eta := \frac{8G}{mn}(1 + \alpha L K).$$

Note that the claim (42) is in fact to show

$$\mathbb{E}_{\mathcal{A}}[\| w^t - \tilde{w}^t \|] \leq \frac{\eta}{\lambda}.$$

This is true for $t = 1$ since $\beta_0 \leq \frac{1}{4L + 2\rho\alpha G} \leq \frac{1}{\lambda}$. Having this, we could easily obtain the result by induction.

3. We are ready to conclude. Note that by Lemma 3, we have

$$\left| \ell \left( w^T - \alpha \nabla \hat{\mathcal{L}}(w^T, \{z_j\}_{j=1}^K), \tilde{z} \right) - \ell \left( \tilde{w}^T - \alpha \nabla \hat{\mathcal{L}}(\tilde{w}^T, \{z_j\}_{j=1}^K), \tilde{z} \right) \right|$$
$$\leq 4G \left\| \left( w^T - \alpha \nabla \hat{\mathcal{L}}(w^T, \{z_j\}_{j=1}^K), \tilde{z} \right) - \left( \tilde{w}^T - \alpha \nabla \hat{\mathcal{L}}(\tilde{w}^T, \{z_j\}_{j=1}^K), \tilde{z} \right) \right\|$$
$$\leq 4\psi(1 + \alpha L) \| w^T - \tilde{w}^T \| \leq 8G \| w^T - \tilde{w}^T \|.$$

Taking expectations from both sides completes the proof for $w^T$. Note that (43) can be extended to $\bar{w}^T$ as well, and using an argument similar to this step, we could show the same stability bound for the average itrtaes as well.

# E Generalization bound for large $K$ regime

Under the premise of Theorem 2, we claim

$$\mathbb{E}_{\mathcal{A},\mathcal{S}}\left[F(w_{\mathcal{S}}) - \hat{F}(w_{\mathcal{S}}, \mathcal{S})\right] \leq \mathcal{O}(1)G^2\left(\frac{1}{mn\mu} + \alpha\min\left\{\frac{LK}{mn\mu}, \frac{1}{\sqrt{K}}\right\}\right). \tag{57}$$

To show this, first, recall that

$$F_i(w) = \mathbb{E}_{\mathcal{D}_i^{\text{test}}}\left[\mathcal{L}_i\left(w - \alpha\nabla\hat{\mathcal{L}}(w, \mathcal{D}_i^{\text{test}})\right)\right].$$

Let $G_i(w) := \mathcal{L}_i\left(w - \alpha\nabla\mathcal{L}_i(w)\right)$. Note that

$$|F_i(w) - G_i(w)| = \left|\mathbb{E}_{\mathcal{D}_i^{\text{test}}}\left[\mathcal{L}_i\left(w - \alpha\nabla\hat{\mathcal{L}}(w, \mathcal{D}_i^{\text{test}})\right) - \mathcal{L}_i\left(w - \alpha\nabla\mathcal{L}_i(w)\right)\right]\right|$$

$$\leq 4\alpha G\mathbb{E}_{\mathcal{D}_i^{\text{test}}}\left|\hat{\mathcal{L}}(w, \mathcal{D}_i^{\text{test}}) - \nabla\mathcal{L}_i(w)\right| \leq 4\alpha\frac{G^2}{\sqrt{K}}.$$

As a result, for $G(w) = \frac{1}{m}\sum_{i=1}^m G_i(w)$, we have

$$|G(w) - F(W)| \leq \mathcal{O}(1)\alpha\frac{G^2}{\sqrt{K}}.$$

Similarly, if we define

$$\hat{G}_i(w) := \hat{\mathcal{L}}\left(w - \alpha\nabla\mathcal{L}_i(w), \mathcal{S}_i^{\text{out}}\right), \quad \hat{G}_w := \frac{1}{m}\sum_{i=1}^m G_i(w),$$

we could show that

$$\mathbb{E}_{\mathcal{S}}\left|\hat{G}(w) - \hat{F}(w, \mathcal{S})\right| \leq \mathcal{O}(1)\alpha\frac{G^2}{\sqrt{K}}.$$

Finally, note that the well-known generalization results for strongly convex functions by using classic stability definition (Definition 1) implies (see [18] for details)

$$\mathbb{E}_{\mathcal{A},\mathcal{S}}|G(w_{\mathcal{A}}) - \hat{G}(w_{\mathcal{A}})| \leq \mathcal{O}(1)\frac{G^2}{mn\mu},$$

where $w_{\mathcal{A}}$ is MAML output. Putting these bounds together, we obtain $\mathcal{O}(1)\left(\frac{G^2}{mn\mu} + \alpha\frac{G^2}{\sqrt{K}}\right)$. Taking minimum of this and Theorem 2 proves the aforementioned claim.

Finally, it is worth mentioning that while we are not sure whether our bound is tight for the large $K$ regime, this is not necessarily the case that the generalization bound improves as $K$ increases. To see this, consider MAML with only one task, i.e., $m = 1$, and the quadratic loss $l(w, z) = (w^\top x - y)^2$ with $z = (x, y)$. In addition, and to focus on the generalization error coming from test update, we assume we have access to exact gradients for outer loop, i.e.,

$$\hat{F}(w) = \frac{1}{\binom{n}{K}}\sum_{\{z_i\}\subset\mathcal{D}^{in}}\mathcal{L}\left(w - \alpha\sum_{i=1}^K\frac{1}{K}\nabla l(w, z_i)\right)$$

Let $\Lambda = \mathbb{E}[xx^\top]$ and $\rho = \mathbb{E}[xy]$. Also, we denote the estimation of $\Lambda$ and $\rho$ over $\mathcal{D}^{in}$ by $\hat{\Lambda}$ and $\hat{\rho}$, respectively.

After some simplifications, it can be shown that

$$\nabla F(w) = \Lambda w - \rho - 2\alpha\Lambda^2 w + 2\alpha\Lambda\rho + \mathcal{O}(\alpha^2),$$

$$\nabla\hat{F}(w) = \Lambda w - \rho + 2\alpha\Lambda\hat{\Lambda}w + \alpha\Lambda\hat{\rho} + \alpha\hat{\Lambda}\rho + \mathcal{O}(\alpha^2).$$

It can be seen that the difference of the two gradients is $\Omega(\frac{\alpha}{n})$ and does not decrease as $K$ increases.

# F Proof of Theorem 3

First, we show the following lemma:

**Lemma 7.** *For any $\tilde{z}$ and any $w \in \mathcal{W}$, we have*

$$\left| \mathbb{E}_{\{z_j^{m+1} \sim p_{m+1}\}_{j=1}^K} \left[ \ell \left( w - \alpha \nabla \hat{\mathcal{L}}(w, \{z_j^{m+1}\}_{j=1}^K), \tilde{z} \right) \right] - \mathbb{E}_{\{z_j^i \sim p_i\}_{j=1}^K} \left[ \ell \left( w - \alpha \nabla \hat{\mathcal{L}}(w, \{z_j^i\}_{j=1}^K), \tilde{z} \right) \right] \right|$$
$$\leq 4\alpha G^2 \|p_{m+1} - p_i\|_{TV}. \tag{58}$$

*Proof.* Note that since $p_i$ are non-atmoic, we could assume $z_j^i$'s are drawn independently. Same story holds for $z_j^{m+1}$'s. Now, for any $j$, let us assume $(z_j^i, z_j^{m+1})$ is drawn from a joint distribution of $p_i$ and $p_{m+1}$ corresponding to the maximal coupling of these distributions, i.e.,

$$z_j^i \sim p_i, \quad z_j^{m+1} \sim p_{m+1}, \quad \mathbb{P}(z_j^i \neq z_j^{m+1}) = \|p_i - p_{m+1}\|_{TV}.$$

Hence, with probability $\binom{K}{t}(\|p_i - p_{m+1}\|_{TV})^t (1 - \|p_i - p_{m+1}\|_{TV})^{K-t}$, we have $z_j^i \neq z_j^{m+1}$ for $t$ choices of $j$ (out of $1, ..., K$).

In addition, similar to the proof of Lemma 4, we could show that

$$\left\| \ell \left( w - \alpha \nabla \hat{\mathcal{L}}(w, \{z_j^{m+1}\}_{j=1}^K), \tilde{z} \right) - \ell \left( w - \alpha \nabla \hat{\mathcal{L}}(w, \{z_j^i\}_{j=1}^K), \tilde{z} \right) \right\|$$
$$\leq 2\alpha G \|\hat{\mathcal{L}}(w, \{z_j^{m+1}\}_{j=1}^K) - \nabla \hat{\mathcal{L}}(w, \{z_j^i\}_{j=1}^K)\|.$$

Hence, if $z_j^i \neq z_j^{m+1}$ for $t$ choices of $j$, then we have

$$\left\| \ell \left( w - \alpha \nabla \hat{\mathcal{L}}(w, \{z_j^{m+1}\}_{j=1}^K), \tilde{z} \right) - \ell \left( w - \alpha \nabla \hat{\mathcal{L}}(w, \{z_j^i\}_{j=1}^K), \tilde{z} \right) \right\|$$
$$\leq 4\alpha G^2 \frac{t}{K}.$$

As a result, we have

$$\mathbb{E}_{\{z_j^{m+1} \sim p_{m+1}\}_{j=1}^K} \left[ \ell \left( w - \alpha \nabla \hat{\mathcal{L}}(w, \{z_j^{m+1}\}_{j=1}^K), \tilde{z} \right) \right] - \mathbb{E}_{\{z_j^i \sim p_i\}_{j=1}^K} \left[ \ell \left( w - \alpha \nabla \hat{\mathcal{L}}(w, \{z_j^i\}_{j=1}^K), \tilde{z} \right) \right]$$
$$\leq \sum_{t=0}^K \binom{K}{t} (\|p_i - p_{m+1}\|_{TV})^t (1 - \|p_i - p_{m+1}\|_{TV})^{K-t} \cdot 4\alpha G^2 \frac{t}{K}$$
$$= 4\alpha G^2 (\|p_i - p_{m+1}\|_{TV}) \sum_{t=0}^K \frac{t}{K} \binom{K}{t} (\|p_i - p_{m+1}\|_{TV})^{t-1} (1 - \|p_i - p_{m+1}\|_{TV})^{K-t}$$
$$= 4\alpha G^2 (\|p_i - p_{m+1}\|_{TV}), \tag{59}$$

where the last equality follows from the fact that

$$\frac{t}{K} \binom{K}{t} (\|p_i - p_{m+1}\|_{TV})^{t-1} (1 - \|p_i - p_{m+1}\|_{TV})^{K-t} = \binom{K-1}{t-1} (\|p_i - p_{m+1}\|_{TV})^{t-1} (1 - \|p_i - p_{m+1}\|_{TV})^{K-1-(t-1)}.$$

$\square$

Let's get back to the proof of Theorem 3. For any $1 \leq i \leq m+1$ and any $\tilde{z}$, let us define

$$X_i(\tilde{z}) := \mathbb{E}_{\{z_j \sim p_i\}_{j=1}^K} \left[ \ell \left( w_{\mathcal{S}} - \alpha \nabla \hat{\mathcal{L}}(w_{\mathcal{S}}, \{z_j\}_{j=1}^K), \tilde{z} \right) \right].$$

In other words, $X_i$ is the loss over data point $\tilde{z}$ when the model is updated using the distribution of task $i$. Next, note that

$$F_{m+1}(w) - F_i(w) = \tag{60}$$
$$\mathbb{E}_{\{z_j \sim p_{m+1}\}_{j=1}^K, \tilde{z} \sim p_{m+1}} \left[ \ell \left( w - \alpha \nabla \hat{\mathcal{L}}(w, \{z_j\}_{j=1}^K), \tilde{z} \right) \right] - \mathbb{E}_{\{z_j \sim p_i\}_{j=1}^K, \tilde{z} \sim p_i} \left[ \ell \left( w - \alpha \nabla \hat{\mathcal{L}}(w, \{z_j\}_{j=1}^K), \tilde{z} \right) \right].$$

Note that by Lemma 4, the term inside expectation is bounded, and hence, by Fubini's theorem, we can cast this term as

$$\mathbb{E}_{\tilde{z}\sim p_{m+1}}[X_{m+1}(\tilde{z})] - \mathbb{E}_{\tilde{z}\sim p_i}[X_i(\tilde{z})] \tag{61}$$

By Lemma 7, we have $|X_i(\tilde{z}) - X_{m+1}(\tilde{z})| \leq 4\alpha G^2 \|p_i - p_{m+1}\|_{TV}$. Hence, we have

$$F_{m+1}(w) - F_i(w) = \mathbb{E}_{\tilde{z}\sim p_{m+1}}[X_{m+1}(\tilde{z})] - \mathbb{E}_{\tilde{z}\sim p_i}[X_i(\tilde{z})] = \mathbb{E}_{\tilde{z}\sim p_{m+1}}[X_{m+1}(\tilde{z})] - \mathbb{E}_{\tilde{z}\sim p_i}[X_{m+1}(\tilde{z})] + e_{i,m},$$

where $|e_{i,m}| \leq 4\alpha G^2 \|p_i - p_{m+1}\|_{TV}$. As a result, we have

$$\left| F_{m+1}(w) - \frac{1}{m}\sum_{i=1}^{m} F_i(w) \right| \leq \left| \mathbb{E}_{\tilde{z}\sim p_{m+1}}[X_{m+1}(\tilde{z})] - \frac{1}{m}\sum_{i=1}^{m} \mathbb{E}_{\tilde{z}\sim p_i}[X_{m+1}(\tilde{z})] \right| + 4\alpha G^2 \|p_i - p_{m+1}\|_{TV} \tag{62}$$

Using Lemma 4, we have $0 \leq X_{m+1}(\tilde{z}) \leq M + 2\alpha G^2$. Hence, by (8), we have

$$\left| \mathbb{E}_{\tilde{z}\sim p_{m+1}}[X_{m+1}(\tilde{z})] - \frac{1}{m}\sum_{i=1}^{m} \mathbb{E}_{\tilde{z}\sim p_i}[X_{m+1}(\tilde{z})] \right| \leq (M + 2\alpha G^2)\|p_{m+1} - \frac{1}{m}\sum_{i=1}^{m} p_i\|_{TV}. \tag{63}$$

Plugging (63) into (62) gives us the desired result.

### F.1 Proof of Corollary 2

Note that

$$\mathbb{E}_{\mathcal{A},\mathcal{S}}[F_{m+1}(w_{\mathcal{S}})] - \min_{\mathcal{W}} F_{m+1} \leq \left( \mathbb{E}_{\mathcal{A},\mathcal{S}}[F_{m+1}(w_{\mathcal{S}}) - F(w_{\mathcal{S}})] \right) + \left( \mathbb{E}_{\mathcal{A},\mathcal{S}}[F(w_{\mathcal{S}})] - \min_{\mathcal{W}} F \right)$$
$$+ \left( \min_{\mathcal{W}} F - \min_{\mathcal{W}} F_{m+1} \right),$$

where the second term on the right hand side is bounded by $\epsilon$ by assumption, and the first and last term are both bounded by $D(p_{m+1}, \{p_i\}_{i=1}^{m})$ based on Theorem 3.

### F.2 Generalization to a task drawn from a distribution of recurring and unseen tasks

Here we show how our result for generalization to an unseen task can be extended to the case that the task at test time is generated from a distribution $\pi$ over both recurring tasks $\{\mathcal{T}_i\}_{i=1}^{m}$ and the unseen task $\mathcal{T}_{m+1}$.

**Corollary 3.** *Under the premise of Theorem 3, and if the task at the test time is generated from the distribution $\pi$ over $\{\mathcal{T}_i\}_{i=1}^{m+1}$, we have*

$$|\mathbb{E}_\pi[F_i(w)] - F(w)| \leq \pi(\mathcal{T}_{m+1})\, D(p_{m+1}, \{p_j\}_{j=1}^{m})(1 - \pi(\mathcal{T}_{m+1}))\sum_{i=1}^{m} |\pi(\mathcal{T}_i) - \frac{1}{m}|\, D(p_i, \{p_j\}_{j=1}^{m}),$$

*where $\pi(\mathcal{T}_i)$ is the probability of task $\mathcal{T}_i$ according to distribution $\pi$.*

*Proof.* Note that

$$|\mathbb{E}_\pi[F_i(w)] - F(w)| \leq \pi_{m+1}|F_{m+1} - F(w)| + (1 - \pi_{m+1})\sum_{i=1}^{m} \pi_i |F_i(w) - F(w)|. \tag{64}$$

Note that by Theorem 3 we have

$$|F_{m+1} - F(w)| \leq D(p_{m+1}, \{p_j\}_{j=1}^{m}), \quad |F_i - F(w)| \leq D(p_i, \{p_j\}_{j=1}^{m}).$$

Plugging these into (64) completes the proof. □

# G  Limitations of the algorithmic stability analysis

Upon reviewers' suggestion, we briefly discuss why the algorithmic stability technique does not lead to meaningful generalization results for nonconvex loss functions. The main issue with applying the stability framework for the nonconvex case is that we have to select a small stepsize to obtain reasonable generalization bounds, but with such small stepsizes, we cannot guarantee that we will find a first-order stationary point (FOSP) solution of the empirical loss in polynomial time.

To be more precise, consider Theorem 3.12 in Section 3.5 of [18]. There, the authors assume the stepsize $\alpha_t$ satisfies the condition $\alpha_t \leq c/t$. To see how this prohibits us from finding an FOSP efficiently, let us recall the convergence analysis of a non-convex smooth objective function $f$. There the main inequality is the following (see Section 1.2.3 in [18]):

$$f(w_T) - f^* \geq \sum_{t=0}^{T} \alpha_t(1 - \alpha_t L/2)\|\nabla f(w_t)\|^2,$$

where $L$ is the smoothness parameter and $w_t$ is $t$-th iterate. It can be shown that by setting the stepsize to $\alpha_t = \Theta(1/t)$, as suggested by [18], we would require $\exp(\Theta(1/\epsilon^2))$ iterations to find an $\epsilon$-FOSP. However, with a constant stepsize, we can achieve the significantly improved rate of $\mathcal{O}(1/\epsilon^2)$ which matches the lower bound for this setting. As this argument shows, to obtain a meaningful generalization bound using algorithmic stability the stepsize should be selected much smaller than the required threshold and as a result the overall iteration/sample complexity could be very large.

Considering this discussion, the algorithmic stability technique imposes a very restrictive assumption on the stepsizes in the *nonconvex setting* which has a detrimental effect on the training error analysis.

# H  A toy example

In this section, we provide a simple numerical experiment to validate our theoretical results. We consider a linear regression problem with dimension $d = 10$ for the case that we have $m$ tasks and $n$ samples per task. For each task $i$, the feature vector $x$ is drawn according to a normal distribution of $\mathcal{N}(\mu_i, 0.2I_d)$, where $\mu_i$ is a vector uniformly at random drawn from $[0, 1]^d$. In addition, for a given $x$, the label $y$ is given by $y = a_i^\top x + \epsilon_i$, where $\epsilon_i \sim \mathcal{N}(0, 0.1)$ and $a_i$ is a random vector. To make tasks similar, we generate the vectors $a_i$ according to $a_i = \frac{u_i + 1_d}{\|u_i + 1_d\|}$, where $u_i$ is a random vector, uniformly drawn from $[0, 1]^d$, and $1_d$ is the all-one vector.

For the loss function, we consider quadratic loss with quadratic regularization, i.e., $l(w, (x, y)) = (w^\top x - y)^2 + \lambda\|w\|^2$, with $\lambda = 0.01$. We choose the number of samples in the stochastic gradient for adaptation as $K = 5$ and the test time learning rate $\alpha = 0.1$, and run MAML for $T = 20000$ iterations.

Figure 2 shows the dependence of test error over recurring tasks on $m$ and $n$. In this case the task at test time is a recurring task. We see that the error decreases as $m$ or $n$ increases which is consistent with our theoretical results.

Next, we consider the case that the task at test time is new and unseen. Note that, in this case, from our theoretical results we know that the error bound includes a term $D(p_{m+1}, \{p_i\}_{i=1}^m)$ which does not decay with $n$. However, if the distributions are close, this term could be relatively small if $m$ is sufficiently large. To study this matter in our example, we consider two cases:

- First, we assume this new task is similar to the observed tasks in training. More formally, similar to the first $m$ tasks, we take $a_{m+1} = \frac{u_{m+1} + 1_d}{\|u_i + 1_d\|}$, where $u_{m+1}$ is again a random vector, uniformly drawn from $[0, 1]^d$. Figure 3 shows the test error in this case. As we expected, here we do not gain that much from increasing $n$, but the error decreases as $m$ increases. This matches our intuition, as for small $m$, i.e., $m = 1$, the distance between two distributions $p_1$ and $p_2$ could be large. However, as $m$ increases, we have tasks where their distributions are close to $p_{m+1}$, and hence the average distance between distributions $p_i, \cdots, p_m$ and $p_{m+1}$ decreases.

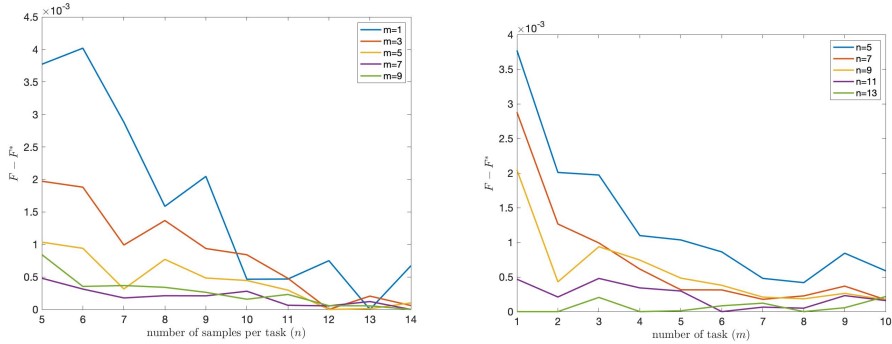

a: Test error as a function of $n$ for different $m$     b: Test error as a function of $m$ for different $n$

Figure 2: Test error over recurring tasks

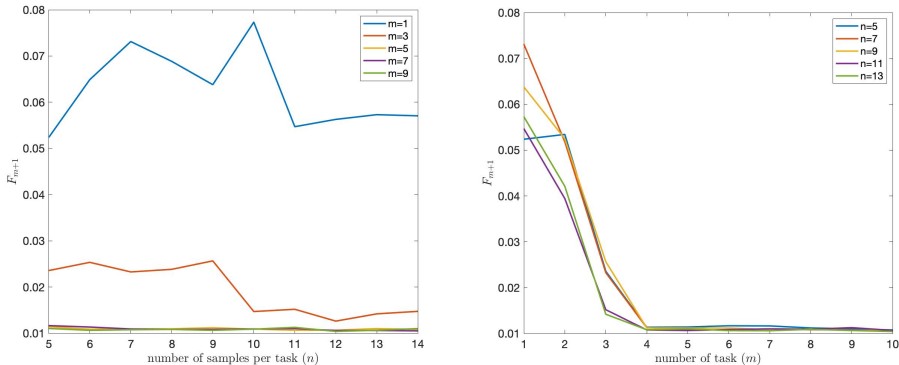

a: Test error as a function of $n$ for different $m$     b: Test error as a function of $m$ for different $n$

Figure 3: Test error over a new but similar task

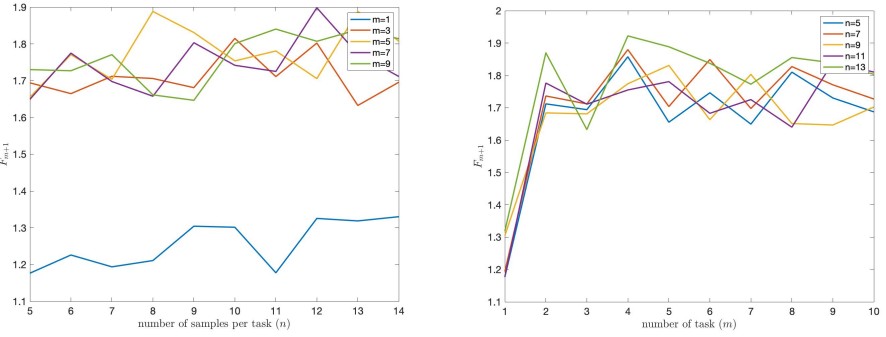

a: Test error as a function of $n$ for different $m$     b: Test error as a function of $m$ for different $n$

Figure 4: Test error over a new and less similar task

- Second, we make this new task less similar to the observed ones. To do so, this time, we choose $a_{m+1} = \frac{u_{m+1} - 1_d}{\|u_i - 1_d\|}$. In this case, we expect to see a relatively large error which does not decrease with either $m$ or $n$, and Figure 4 exactly shows this matter.