# OpenReview forum: "Generalization of Model-Agnostic Meta-Learning Algorithms: Recurring and Unseen Tasks"
_NeurIPS.cc/2021/Conference — NeurIPS 2021 Poster_

### Official Review · Reviewer_aLZn · 2021-07-15

**Rating:** 4
**Confidence:** 4

**Summary:**

This paper studies the generalization properties of MAML in the strongly convex setting. It's assumed that there are $m$ training tasks, each with $n$ samples, and the inner loop adaptation is one step of SGD using $K$ samples. It's proved that if the new task is a uniformly random task from one of the $m$ training tasks, MAML incurs a generalization error of $O(\frac{K}{mn})$; if the new task is an unseen task, the error depends on the TV distance between the new task distribution and all the training task distributions as well as their average. The analysis is based on a modified version of algorithmic stability tailored to the MAML setup.

**Limitations And Societal Impact:**

As noted in the main review, I believe the generalization result to an unseen task is very loose and can be misleading. A discussion of this is needed.

**Main Review:**

Originality: The contribution of this paper is new. The paper obtains an $O(1/mn)$ bound in the strongly convex setting, while the previous result in [25] is $O(1/m)$ (though [25] covers non-convex losses). The proof technique is based on algorithmic stability but also needs nontrivial modification.

Quality: The paper is technically sound. The results are clearly stated and discussed.

Clarity: The paper is clearly written and well organized.

Significance:
- I think the most interesting result is the $O(1/mn)$ error bound for "in-task generalization", i.e., generalizing to a random training task. The previous bound for MAML is only $O(1/n)$ in [25]. To obtain the $O(1/mn)$ bound, a new type of stability (Definition 3) is required, which is non-trivial.
- However, I believe the result of generalization to an unseen task (Theorem 3) is very loose and does not capture the actual interesting aspects of meta-learning. Essentially, for this bound to be small it requires the new task distribution to be very close in TV distance to all the training tasks as well as their average. However, this is almost never the case. For example, suppose that all the tasks are drawn i.i.d. from a meta-distribution over tasks (which is usually true in a lot of few-shot learning benchmarks). Then $p_1, p_2, \ldots, p_{m+1}$ could all be very different from each other, but we can still expect MAML to generalize well to $p_{m+1}$, since it's a task drawn from the same meta-distribution. The bound provided in the current paper dramatically fails to capture this.
- The strong convexity is certainly very restrictive, and it's noted that the technique used in the paper cannot generalize beyond the convex case. That said, this is understandable and acceptable as a first step for theoretical purpose.

Other questions:
- The paper only considers 1-step SGD adaptation in the inner loop. Is there any hope to generalize the result to multiple steps?
- It seems that a somewhat non-standard data splitting procedure is used, i.e. first splitting the data into $S_i^{in}$ and $S_i^{out}$ each with size $n$, and then the samples used for inner and outer loops are sampled from these two subsets. How important is this in the theoretical analysis and is it unavoidable?

**Time Spent Reviewing:**

4

---

> ### Author Response · Authors · 2021-08-11
> **Response to Reviewer aLZn (Part 1)**
>
> We thank the reviewer for the constructive feedback. In the following paragraphs we briefly address the issues raised by the reviewer.
>
> **Comment:** However, I believe the result of generalization to an unseen task (Theorem 3) is very loose and does not capture the actual interesting aspects of meta-learning. Essentially, for this bound to be small it requires the new task distribution to be very close in TV distance to all the training tasks as well as their average. However, this is almost never the case. For example, suppose that all the tasks are drawn i.i.d. from a meta-distribution over tasks (which is usually true in a lot of few-shot learning benchmarks). Then $p_1, p_2,...,p_{m+1}$ could all be very different from each other, but we can still expect MAML to generalize well to $p_{m+1}$, since it's a task drawn from the same meta-distribution. The bound provided in the current paper dramatically fails to capture this.
>
> **Response:**
>  We thank the reviewer for this great question. We believe our bounds are not loose. Next, we explain the reason that we believe that the bound in Theorem 3 is tight. For the ease of this discussion, let us focus on the case that we have access to the exact gradient.
>
> First, recall that our bound  in Theorem 3 has two main terms (we suppress the constants to simplify the notation):
>
> - The first term is $||p\_{m+1} - \frac{1}{m} \sum\_{i=1}^m p_i ||\_{TV}$. This term comes from the generalization error of the classic ERM problem, and it persists even with $\alpha=0$. In fact, we believe having this term is inevitable. To see this, let us focus on the case that $\alpha=0$ and we don't have any adaptation step. In such setting, the MAML formulation boils down to an ERM problem with $m$ tasks with distributions $p_1$,..., $p_m$. Since $\alpha=0$, the ERM problem in this case is equivalent to only having one task with distribution $\bar{p} := \frac{1}{m} \sum_{i=1}^m p_i$. So now, it's like we train with respect to $\bar{p}$ and test on $p\_{m+1}$, and the error of doing so is bounded by $|| \bar{p} - p\_{m+1}||\_{TV}$ by total variation distance definition (Definition 4).
>
>
> - The other term is $\alpha \frac{1}{m} \sum\_{i=1}^m ||p\_{m+1} - p\_i ||\_{TV}$. Note that this bound is for fixed $p_1,...,p\_{m+1}$, and we do not assume that these $p\_i$'s are drawn from an underlying distribution. But let us see what we obtain if we assume the $p\_i$'s are drawn from a common distribution $\mathcal{P}$ over a family of distributions. In this case, $\alpha \frac{1}{m} \sum\_{i=1}^m ||p\_{m+1} - p\_i ||\_{TV}$ in expectation is equal to $\alpha \mathbb{E}\_{q\_1,q\_2 \sim \mathcal{P}} ||q\_1 - q\_2 ||$ which is a constant that does not go to zero even for large $m$. This might seem loose, however, this is not the case.
> Let us elaborate this matter with an example.  Consider the loss function $l(w,(x,y)) = \frac{1}{2} (w^\top x - y)^2$. In this case, it is straightforward to see that the expected loss of task $i$ is given by
>
>  $$ L_i(w) = E_{(x,y)\sim p_i} [l(w,(x,y))] = \frac{1}{2} w^\top \Sigma_i w - w^\top \rho_i + \sigma_i^2,$$
>
>  where $\Sigma_i = \mathbb{E}\_{(x,y) \sim p\_i}[x x^\top]$, $\rho\_i = \mathbb{E}\_{(x,y) \sim p\_i}[xy]$, and $\sigma_i^2 = \mathbb{E}\_{(x,y) \sim p\_i}[y y^2]$. Let us assume $\Sigma\_i = \gamma\_i I_d$ where $\gamma\_i$ is drawn from a uniform distribution over $(1-\zeta, 1+\zeta)$. Also, we assume $\rho_i$ is drawn from an independent distribution with mean $\bar{\rho}$. Let $w^*$ be the solution of the MAML problem
>
>  $$ \min\_{w}\frac{1}{m} \sum\_{i=1}^m L_i(w - \alpha \nabla L\_i(w)).$$
>
>  From [13], we know that
>
>  $$w^* = (\frac{1}{m} \sum_{i=1}^m (I - \alpha \Sigma_i)^2 \Sigma_i)^{-1} (\frac{1}{m} \sum\_{i=1}^m (I - \alpha \Sigma_i)^2 \rho_i).$$
>
> We can see that for large $m$ the first term on the right hand side concentrates around
>
> $$\mathbb{E}\_{\Sigma}[((I - \alpha \Sigma)^2 \Sigma)^{-1}] $$
>
> and the second term concentrates around
>
> $$\mathbb{E}\_{\Sigma, \rho}[(I - \alpha \Sigma)^2 \rho].$$
>
> Ignoring terms of order $\mathcal{O}(\alpha^2)$, and using the fact that $\mathbb{E}[\Sigma^2] = (1 + \zeta^2/3) I_d$, we can see that for large $m$ we approximately have
>
> $$w^* = \frac{1 - 2\alpha}{1-2\alpha - 2/3 \alpha \zeta^2} \bar{\rho}.$$
>
> Now, we look at $\mathbb{E}\_{\Sigma\_{m+1}, \rho\_{m+1}} [|| \nabla F\_{m+1}(w^*) ||^2]$. Note that if we show this is greater than a non-vanishing term, then the excess error will have a non-vanishing term as well, since due to strong convexity we can translate this bound to excess risk.
>
> To show this, it suffices to lower bound $|| \mathbb{E}\_{\Sigma\_{m+1}, \rho\_{m+1}} [\nabla F\_{m+1}(w^*)] ||$. Note that
>
> $$ \nabla F\_{m+1}(w^*) = (I - \alpha \Sigma\_{m+1})^2 (\Sigma\_{m+1} w^* - \rho\_{m+1}),$$
>
> and hence for small $\alpha$ we have
>
> $$ || \nabla F\_{m+1}(w^*) || \geq (1 - \alpha (1+\zeta))^2  || \Sigma\_{m+1} w^* - \rho\_{m+1} ||.$$
>
> This implies that
>
> $$ \mathbb{E} [ || \nabla F\_{m+1}(w^*) || ] \geq (1 - \alpha (1+\zeta))^2 \mathbb{E} [ || \Sigma\_{m+1} w^* - \rho\_{m+1} ||]$$
> $$ \geq (1 - \alpha (1+\zeta))^2 || \mathbb{E}[\Sigma\_{m+1} w^* - \rho\_{m+1}] || $$
> $$ = (1 - \alpha (1+\zeta))^2 || w^* - \bar{\rho} ||$$
> $$ ~= (1 - \alpha (1+\zeta))^2 (1-  \frac{1 - 2\alpha}{1-2\alpha - 2/3 \alpha \zeta^2}) ||\bar{\rho}||$$
>
> which is a non-vanishing term.
>
> We hope this shows that the bound is not loose for an unseen task, and we would be happy to provide further clarification on this matter. As a final note, it is worth answering this question that if MAML adds this second term to the generalization error compared to ERM, then why it performs better. Note that the baseline of the excess risk is different for both methods, and the baseline for MAML is smaller as it looks at the error after updating the model. Hence, when distributions are close, i.e., $\alpha \mathbb{E}_{q_1,q_2 \sim \mathcal{P}} || q_1 - q_2 ||$ is small, we benefit from MAML.

---

> > ### Author Response · Authors · 2021-08-11
> > **Response to Reviewer aLZn (Part 2)**
> >
> > **Comment:** The strong convexity is certainly very restrictive, and it's noted that the technique used in the paper cannot generalize beyond the convex case. That said, this is understandable and acceptable as a first step for theoretical purpose.
> >
> > **Response:**
> > Thanks for the constructive feedback. Here, we briefly explain why the algorithmic stability technique cannot handle the nonconvex case as it leads to a significantly suboptimal training error:
> >
> > Consider Theorem 3.12 in Section 3.5 of [Hardt et al., 2016] where a generalization bound using algorithmic stability technique is provided for nonconvex settings. There, the authors assume the stepsize $\alpha_t$ satisfies the condition $\alpha_t \leq c/t$. To see how this prohibits us from finding a first-order stationary point (FOSP) efficiently, let us recall the convergence analysis of a non-convex smooth objective function $f$. There the main inequality is the following (see Section 1.2.3 in [33]):
> >
> > $$f(w_T) - f^* \geq \sum_{t=0}^T \alpha_t (1-\alpha_t L/2) \| \nabla f(w_t) \|^2,$$
> > where $L$ is the smoothness parameter and $w_t$ is $t$-th iterate. It can be shown that by setting the stepsize to $\alpha_t = \Theta(1/t)$, as suggested by [Hardt et al., 2016], we would require  $\exp(\Theta(1/\epsilon^2))$ iterations to find an $\epsilon$-FOSP. However, with a constant stepsize, we can achieve the significantly improved rate of $\mathcal{O}(1/\epsilon^2)$ which matches the lower bound for this setting. As this argument shows, to obtain a meaningful generalization bound using algorithmic stability the stepsize should be selected much smaller than the required threshold and as a result the overall iteration/sample complexity could be very large.
> >
> > Considering this discussion, the algorithmic stability technique imposes a very restrictive assumption on the stepsizes in the *nonconvex setting* which has a detrimental effect on the training error analysis.
> >
> > $~$
> >
> > **Comment:** The paper only considers 1-step SGD adaptation in the inner loop. Is there any hope to generalize the result to multiple steps?
> >
> > **Response:** Yes. We believe the framework that we propose can be extended to multi-step MAML, i.e., the case that we perform more than one step of stochastic gradient descent at test time.
> > To extend our work to that case, we basically need to adapt the $(\gamma, K)$-uniform stability definition, i.e., Definition 3, accordingly. To be more precise, let us define the following function
> > $\Psi(w, \mathcal{D}) = w - \alpha \nabla \hat{L}(w,\mathcal{D})$.
> > Considering this definition we extend our notion of algorithmic stability to MAML with multiple steps of adaptation. For instance, for MAML with two steps of gradient updates, the corresponding definition would be the following:
> >
> >
> > **Definition** A randomized algorithm $A$ with output $w_S$ given dataset $S$ is called $(\gamma, K)$-uniformly stable if the following condition holds for any $i \in \{1,\ldots m\}$: Take the dataset $\tilde{S}$ which is the same as $S$, except that $\tilde{S}\_i^{in}$ and $\tilde{S}\_i^{out}$ differ from $S\_i^{in}$ and $S\_i^{out}$ in at most $2K$ and one data points, respectively. Then, for any $\tilde{z} \in \mathcal{Z}$ and any $2K$ distinct points $\{z_1,...,z_{2K}\}$ in $\mathcal{Z}$,
> >
> > $$
> > \mathbb{E}\_A [ |\ell (\Psi(\Psi(w\_S, (z_j)\_{j=1}^K), (z_j)\_{j=K+1}^{2K}) , \tilde{z} ) - \ell (\Psi(\Psi(w\_{\tilde{S}}, (z\_j)\_{j=1}^K), \(z\_j)\_{j=K+1}^{2K}) , \tilde{z} ) | ] \leq \gamma.
> > $$
> >
> >
> > The next step would be extending Theorem 1 to this definition. Going over the proof of the current version of Theorem 1, one can see that the extension to this definition is pretty straightforward. The last step would be characterizing the stability parameter of MAML with respect to this definition. We will include this extension and the corresponding result as a separate section to the appendix. Thanks for your suggestion.
> >
> > $~$
> >
> > **Comment:** It seems that a somewhat non-standard data splitting procedure is used, i.e. first splitting the data into $S_i^{in}$ and $S_i^{out}$ each with size $n$, and then the samples used for inner and outer loops are sampled from these two subsets. How important is this in the theoretical analysis and is it unavoidable?
> >
> > **Response:** Thanks for the great question. We mainly need this separation for the proof of Theorem 1. To be more precise, consider the equation just after line 565 in the proof of Theorem 1. There, we argue that all expectations
> > of the form
> > $\mathbb{E}\_{A, S} \left [\ell \left (w\_S - \alpha \nabla \hat{L}(w_S,(z^{in}_{\zeta_j})\_{j=1}^K), z^{out}\_{\tilde{\zeta}} \right) \right ]$ that appear on the right hand side of that expression are equal to each other due to symmetry.
> >
> > If we don't use such splitting, this argument does not hold, since in that case $z^{out}\_{\tilde{\zeta}}$ could be among the $K$ data points of $(z^{in}\_{\zeta_j})\_{j=1}^K$. In other words, the symmetry will break in that case.
> >
> > We decided to use the described splitting approach to simplify our analysis. However, we believe it should be still possible to derive generalization bounds without splitting the dataset. This can be done by treating the cases that $z^{out}_{\tilde{\zeta}}$ belongs to $(z^{in}\_{\zeta_j})\_{j=1}^K$ as a bad event with a small probability that leads to an additional error term that can be bounded.

---

> > > ### Comment · Reviewer_aLZn · 2021-08-24
> > > **Response from reviewer**
> > >
> > > Many thanks to the authors for their detailed response.
> > >
> > > I'm still not very convinced by the argument regarding Theorem 3. It seems that in the iid tasks example provided in the response, the problem is inherently difficult if we are only allowed to do one step of SGD using a sufficiently small step size, because different tasks have different optimal solutions. What if we consider a more benign scenario where all the tasks have their optimal solutions close to a certain $w^*$, but they have very different input marginal distributions? Then it seems that with sufficiently many tasks, you can learn something close to $w^*$, which will lead to good performance on an unseen task. However the TV distance between these distributions will still be large.

---

> > > > ### Author Response · Authors · 2021-08-31
> > > > **Response to Further Comments by Reviewer aLZn**
> > > >
> > > > Thank you for your comments and feedback. We agree with the reviewer that our bound is for the general case and might not be optimal in certain special cases, but we believe our framework can be used to derive bounds for those special cases as well. For instance, considering reviewer's example, let us assume all tasks' optimal solutions are within distance $R$ of each other. We provide a sketch of how we could derive a generalization bound in this case, and we will include a more detailed and rigorous version of this proof in our final manuscript.
> > > >
> > > > Let $w\_\mathcal{S}$ be the output of the algorithm, and let $w^\ast\_F$ and $w^\ast\_{new}$ be the optimal solutions of $F$ and the new task, respectively. The generalization result allows us to bound $\|w\_\mathcal{S} - w^\ast\_F\|$ by $\mathcal{O}(\frac{1}{mn})$. We could also use the above assumption to bound $\|w^\ast\_F - w^\ast_{new}\|$ by a function of $R$ (let's use $R$ itself as the bound for simplicity). Hence, we will have $\|w\_\mathcal{S} - w^\ast\_{new}\| \leq \mathcal{O}( \frac{1}{mn} + R)$, and we can easily translate it to function value difference using smoothness property.

---

### Official Review · Reviewer_xQuq · 2021-07-16

**Rating:** 6
**Confidence:** 3

**Summary:**

The paper analyzes the generalization error of MAML in 2 scenarios that are different in the new tasks at test time which are (i) the training tasks and (ii) unseen tasks, under the assumptions that the loss function is (strongly) convex and the number of steps of SGD in the adaptation step of MAML is 1. In the former case, this work proposes a new notion of stability that is capable of capturing the dependence on the number of data points per task in the bound. In the latter case, this work formulates a generalization error bound based on the total variation distance between the distribution of the new task and those of the training task.

**Main Review:**

(originality & significance) I think that the paper presents an improved result over the existing work and other new theoretical results. In particular, it proposes a new notion of stability to construct a generalization error bound that depends on the number of data points per task, which allows a tighter bound compared with an existing work. It also proposes a new approach of using the total variation distance to analyze the generalization bound when the task at test time does not come from the training tasks.

(quality):  The main weaknesses of the analysis is in the assumption of the convex loss function because MAML is often used to learn a deep neural network whose loss is non-convex. Therefore, the implications of such analysis to the real applications of MAML remain unknown. However, I acknowledge that the paper justifies this weakness in Remark 2. Another limitation is about the restriction of the number of SGD steps in the adaptation of MAML. The paper could be improved by mentioning the implications (or challenges) when the number of SGD steps in the adaptation of MAML is larger than 1.

The paper is generally well-structured and well-written. There are only very few minor typos:
* Assumption 1(ii): “unbounded”.
* Line 196: “from [1]” should be from [24] if I’m not wrong.
* Line 218: “deferentially private” should be “differentially private”.
* Lines 286 and 287: “k” should be “K”.
* Title of section 3.3: “Task” is capitalized.


**Time Spent Reviewing:**

5

---

> ### Author Response · Authors · 2021-08-11
> **Response to Reviewer xQuq**
>
> We thank the reviewer for the constructive feedback. In the following paragraphs we briefly address the issues raised by the reviewer.
>
> **Comment:** The main weakness of the analysis is in the assumption of the convex loss function because MAML is often used to learn a deep neural network whose loss is non-convex. Therefore, the implications of such analysis to the real applications of MAML remain unknown. However, I acknowledge that the paper justifies this weakness in Remark 2.
>
> **Response:**
>  We agree with the reviewer that our assumptions do not hold for neural networks in general nonconvex functions. However, as we have stated in Remark 2, the main reason that we have imposed the strongly convex assumption is that the algorithmic stability technique for deriving generalization bounds is mainly limited to convex settings and its application to nonconvex settings leads to suboptimal training error. Next, we briefly explain this issue.
>
> Consider Theorem 3.12 in Section 3.5 of [Hardt et al., 2016] where a generalization bound using algorithmic stability technique is provided for nonconvex settings. There, the authors assume the stepsize $\alpha_t$ satisfies the condition $\alpha_t \leq c/t$. To see how this prohibits us from finding a first-order stationary point (FOSP) efficiently, let us recall the convergence analysis of a non-convex smooth objective function $f$. There the main inequality is the following (see Section 1.2.3 in [33]):
>
> $$f(w_T) - f^* \geq \sum_{t=0}^T \alpha_t (1-\alpha_t L/2) \| \nabla f(w_t) \|^2,$$
> where $L$ is the smoothness parameter and $w_t$ is $t$-th iterate. It can be shown that by setting the stepsize to $\alpha_t = \Theta(1/t)$, as suggested by [Hardt et al., 2016], we would require  $\exp(\Theta(1/\epsilon^2))$ iterations to find an $\epsilon$-FOSP. However, with a constant stepsize, we can achieve the significantly improved rate of $\mathcal{O}(1/\epsilon^2)$ which matches the lower bound for this setting. As this argument shows, to obtain a meaningful generalization bound using algorithmic stability the stepsize should be selected much smaller than the required threshold and as a result the overall iteration/sample complexity could be very large.
>
> Considering this discussion, the algorithmic stability technique imposes a very restrictive assumption on the stepsizes in the *nonconvex setting* which has a detrimental effect on the training error analysis.
>
> $~$
>
> **Comment:**
> Another limitation is about the restriction of the number of SGD steps in the adaptation of MAML. The paper could be improved by mentioning the implications (or challenges) when the number of SGD steps in the adaptation of MAML is larger than 1.
>
>
> **Response:**
> We believe the framework that we propose can be extended to multi-step MAML, i.e., the case that we perform more than one step of stochastic gradient descent at test time.
> To extend our work to that case, we basically need to adapt the $(\gamma, K)$-uniform stability definition, i.e., Definition 3, accordingly. To be more precise, let us define the following function
> $\Psi(w, \mathcal{D}) = w - \alpha \nabla \hat{L}(w,\mathcal{D})$.
> Considering this definition we extend our notion of algorithmic stability to MAML with multiple steps of adaptation. For instance, for MAML with two steps of gradient updates, the corresponding definition would be the following:
>
>
> **Definition** A randomized algorithm $A$ with output $w_S$ given dataset $S$ is called $(\gamma, K)$-uniformly stable if the following condition holds for any $i \in \{1,\ldots m\}$: Take the dataset $\tilde{S}$ which is the same as $S$, except that $\tilde{S}\_i^{in}$ and $\tilde{S}\_i^{out}$ differ from $S\_i^{in}$ and $S\_i^{out}$ in at most $2K$ and one data points, respectively. Then, for any $\tilde{z} \in \mathcal{Z}$ and any $2K$ distinct points $\{z_1,...,z_{2K}\}$ in $\mathcal{Z}$,
>
> $$
> \mathbb{E}\_A [ |\ell (\Psi(\Psi(w\_S, (z_j)\_{j=1}^K), (z_j)\_{j=K+1}^{2K}) , \tilde{z} ) - \ell (\Psi(\Psi(w\_{\tilde{S}}, (z\_j)\_{j=1}^K), \(z\_j)\_{j=K+1}^{2K}) , \tilde{z} ) | ] \leq \gamma.
> $$
>
> The next step would be extending Theorem 1 to this definition. Going over the proof of the current version of Theorem 1, one can see that the extension to this definition is pretty straightforward. The last step would be characterizing the stability parameter of MAML with respect to this definition. We will include this extension and the corresponding result as a separate section to the appendix. Thanks for your suggestion.
>
> $~$
>
> **Comment:** The paper is generally well-structured and well-written. There are only very few minor typos:
>
> **Response:**
>  Thank you for catching these typos. We will fix them in the revised version.

---

### Official Review · Reviewer_xD4r · 2021-07-17

**Rating:** 7
**Confidence:** 4

**Summary:**

The authors present generalization bounds for MAML when test tasks are the same as the ones seen during training (recurring tasks) or when they are different but sampled from the same distribution (unseen tasks). The main takeaways are a. for recurring tasks, the excess population loss is bounded by O(1/mn) with m the number of tasks and n the number of data points, and b. generalization to unseen tasks depends on the distance (total variation) between train and test task distributions.

**Limitations And Societal Impact:**

Limitations are clearly stated, societal impact is not mentioned.

**Main Review:**

The results of the paper are novel and of interest to the multi-task/meta-learning community. In particular, the generalization result for recurring tasks (Th. 1 & 2) refines multi-tasks bounds for MAML when data is limited. This is important since limited data is often the main motivation for these types of meta-learning methods. While the approach to get those bounds builds on the stability of SGD, the proposed stability definition is tailored to MAML and lends itself to further study and extensions. The second result (Th. 3) directly targets transfer to new tasks, formalizing one of the main intuitions behind MAML: the closer the train and test tasks, the most effective the transfer. To the best of my knowledge, this is the first work directly assessing generalization for MAML when train and test task sets are disjoint.

It helps that the paper is clearly written and introduces all necessary background to MAML. For each theorem, assumptions are clearly stated and the results are discussed from both a theoretical and practical perspective. Although a bit dense, I enjoyed reading this paper.

The main limitation of the paper is convexity of the meta-learning loss — but similar publications also considered this setting. Specifically, I would like to see the assumptions on the fast-adaptation learning rate (alpha) relaxed, since they are rarely met in practice.

Finally, I wished the related work section were a bit more expansive. For example, issues of generalization were also considered in [1], and [2] discussed issues related to the number of samples per tasks. How does the present work differ?

References:

[1] "Global Convergence and Generalization Bound of Gradient-Based Meta-Learning with Deep Neural Nets", Wang et al., ArXiv 2020.

[2] "How to distribute data across tasks for meta-learning?", Cioba et al., ArXiv 2021.

**Time Spent Reviewing:**

7

---

> ### Author Response · Authors · 2021-08-11
> **Response to Reviewer xD4r**
>
> We thank the reviewer for the constructive feedback. In the following paragraphs we briefly address the issues raised by the reviewer.
>
> **Comment:** The main limitation of the paper is convexity of the meta-learning loss — but similar publications also considered this setting. Specifically, I would like to see the assumptions on the fast-adaptation learning rate (alpha) relaxed, since they are rarely met in practice.
>
> **Response:** We agree with the reviewer that the convexity assumption is the main limitation of our work. We see studying the nonconvex case as a future direction, although, as we have discusses in Section 4, this could be challenging, since the generalization of gradient methods is not well understood in the nonconvex setting even for the classic supervised learning problem. Please also see our response to Reviewer acnS regarding the limitations of the algorithmic stability scheme for nonconvex settings.
>
> Regarding the assumption on the fast-adaptation learning rate $\alpha$, we should add that this assumption is mainly needed to certify the convexity of the meta-objective function. If we relax this assumption, the meta-objective could become nonconvex and our framework would not be applicable.
>
> $~$
>
> **Comment:**
> Finally, I wished the related work section were a bit more expansive. For example, issues of generalization were also considered in [1], and [2] discussed issues related to the number of samples per tasks. How does the present work differ?
>
> References:
>
> [1] "Global Convergence and Generalization Bound of Gradient-Based Meta-Learning with Deep Neural Nets", Wang et al., ArXiv 2020.
>
> [2] "How to distribute data across tasks for meta-learning?", Cioba et al., ArXiv 2021.
>
> **Response:**
>  We thank the reviewer for bringing these two papers to our attention, and we will make sure to discuss both of these papers in our revised manuscript. Next, we briefly mention the differences between these two papers and our submission:
>
>  - The generalization result in the first paper (Wang et al.) is obtained under the assumption that we have access to new samples at each iteration (please see Algorithm 2 in that paper) while our work allows multi-pass over the data which is the common implementation of machine learning algorithms in practice. Note that studying generalization when we have access to new samples is easier since the gradients are independent from previous iterations, and hence we can directly obtain bound on the population loss, and we don't need to even break down the error to generalization and training errors and use techniques such as stability or Rademacher complexity to bound the generalization error. Please also check lines 33-43 in the introduction in this regard.
>
>  - The second paper also mainly focuses on the mixed linear regression case where the generated data and the underlying tasks' parameters are all coming from Gaussian distributions. In that case, due to this specific structure of the problem, the test error can be characterized explicitly. However, in our paper, we consider a much more general case, and we need to develop bounds using techniques such as algorithmic stability, since we cannot derive the exact solution.
>
>  We will highlight these points in the literature review of the revised paper.

---

> > ### Comment · Reviewer_xD4r · 2021-08-30
> > **Reply**
> >
> > Thank you for the detailed reply; I keep my positive assessment.

---

### Official Review · Reviewer_CKWF · 2021-07-18

**Rating:** 6
**Confidence:** 4

**Summary:**

In this paper, the authors provide a generalization and uniform-stability-type analysis for MAML training. In specific, the authors first present a generalization error bound (based on the uniform stability bound) of $O(1/mn)$ for strongly convex objective functions when the new task at the test time belong to one of the training tasks. Then, for a more interesting setting where the new task does not belong to the training tasks, the authors provide a generalization bound consisting of a total variation distance between the data distributions of this unseen task and the training tasks. One toy experiment is provided in the supplementary materials to verify the impact of the sizes $m,n$ on the generalization performance.

**Limitations And Societal Impact:**

Yes

**Main Review:**

In this paper, the authors provide a generalization and uniform-stability-type analysis for MAML training. In specific, the authors first present a generalization error bound (based on the uniform stability bound) of $O(1/mn)$ for strongly convex objective functions when the new task at the test time belong to one of the training tasks. Then, for a more interesting setting where the new task does not belong to the training tasks, the authors provide a generalization bound consisting of a total variation distance between the data distributions of this unseen task and the training tasks. One toy experiment is provided in the supplementary materials to verify the impact of the sizes $m,n$ on the generalization performance. My detailed comments are given as below.

Pros:
1.	This paper is well written and easy to follow. I find that the notations, definitions, assumptions and related works in this paper are clear and the explanation for the main results are clear and reasonable.
2.	Studying the generalization for MAML type of algorithms is very important, and this paper takes a good try to provide generalization error and stability bounds for MAML under various settings.
3.	This paper shows that the sample number for the intra-task training is also important and can be used to further reduce the generalization error.

Cons:
1.	My first concern is the tightness of the upper bound. It is true that the dependence on $mn$ (i.e., the total number of training sample) is optimal for the strongly convex objective function (in fact, this is a  standard result in statistical learning where the optimal dependence of the excess risk on the sample size M is $O(M)$). However, it is not clear whether other parameters (even some constants) are still optimal for the authors’ upper bounds, e.g., the authors do not prove the optimality of the test sample size $K$. In addition, it seems to me that the results in this paper are not surprising (even though the derivations can be different due to the MAML structure), i.e., the bounds decrease with the total number $mn$ of the training samples (as we always see in standard supervised learning). I will be more excited if the derived upper bounds can involve some special components caused by the composite structure of MAML or the upper bounds capture some interplay between the training and the test procedures of the meta training process of MAML. In Chen et al., 2020 (as pointed out by the authors), they provide interesting findings on how the support/query split used in the meta learning (including MAML) affects the generalization (as well as the theoretical generalization bounds). However, I do not see much new things for MAML in the generalization bounds in this paper.
2.	My another concern is that the empirical verification is not sufficient. First, the curves in the experiments contain  large variations, and I suggest the authors can run more seeds and do an average to make the results convincing. Second, the authors only verify the impact of sample sizes $m,n$ on the generalization performance. Since the sample size $K$ of the test data is also important, I suggest an experiment to test its impact on the generalization. Third, the authors provide a generalization bound for an unseen task, which increases with the total variation distance between tasks’ distributions. Therefore, it would be better if the authors can provide experiments to verify whether such a total variation distance metric really captures the generalization error. If so, this will be a good contribution because it may provide guidelines for how to improve the generalization of meta-learning based on the task selection.  Fourth, since the supervised classification is an important application for meta-learning, it would be better if an experiment can be done for it.
3.	The  training error characterization mainly follows from  exiting works e.g., Fallah et al., 2020, and the stability analysis is somewhat based on Bousquet & Elisseeff, 2020. Can the authors clarify here what is challenging when extending to the MAML training?

In summary, I think this paper takes a good step to study the generalization of MAML, and it involves some novel characterizations, e.g., eliminating the dependence on $K$ for the total variation distance characterization via the maximum coupling in Hollander, 2012. Although the derivations in this paper may contain new treatments, I do not find the results interesting or exciting and the experiments are not sufficient to verify the generalization of MAML in broader applications. To attract more people form this meta-learning area,  I feel these bounds are not sufficient and more efforts should be taken, e.g., to show that such total variation bound has a tight impact on the generalization of MAML. Therefore, I tend to weakly reject this paper, but I am open to increase my score based on the authors’ feedback and other reviewers’ comments.

**Time Spent Reviewing:**

4

---

> ### Author Response · Authors · 2021-08-11
> **Response to Reviewer CKWF**
>
> We thank the reviewer for the constructive feedback. In the following paragraphs we briefly address the issues raised by the reviewer.
>
> **Comment:**
> My first concern is the tightness of the upper bound. It is true that the dependence on $mn$
>  (i.e., the total number of training sample) is optimal for the strongly convex objective function (in fact, this is a standard result in statistical learning where the optimal dependence of the excess risk on the sample size M is $O(M)$). However, it is not clear whether other parameters (even some constants) are still optimal for the authors’ upper bounds, e.g., the authors do not prove the optimality of the test sample size $K$. In addition, it seems to me that the results in this paper are not surprising (even though the derivations can be different due to the MAML structure), i.e., the bounds decrease with the total number $mn$ of the training samples (as we always see in standard supervised learning). I will be more excited if the derived upper bounds can involve some special components caused by the composite structure of MAML or the upper bounds capture some interplay between the training and the test procedures of the meta training process of MAML. In Chen et al., 2020 (as pointed out by the authors), they provide interesting findings on how the support/query split used in the meta learning (including MAML) affects the generalization (as well as the theoretical generalization bounds). However, I do not see much new things for MAML in the generalization bounds in this paper.
>
> **Response:**
>  While this is true that with $M$ samples, we *expect* to get an error of $O(1/M)$, this does not immediately follow for the MAML-type algorithms from the existing results and techniques in the literature. In fact, as discussed in Remark 1, the existing results in the literature will give us the bound $O(1/m)$ instead of $O(1/mn)$.
>
>  Regarding the tightness of our bound, it is worth noting that the classic lower bound for SGD over strongly convex functions translates to a $\mathcal{O}(1/mn)$ lower bound in our setting, which matches the bound that we have established for the regime that $K$ is small. Therefore, our generalization bound is tight for the case that $K$ is small (relative to $n$). We should add that this setting is often the case for most real-world applications of  meta-learning, such as $K$-shot learning. In addition, we provide a different bound in Appendix E which is tighter than Theorem 2's bound for the large $K$ regime. We agree with the reviewer that it might be possible to establish a tighter bound for the large $K$ setting.
>
> $~$
>
> **Comment:**
>  The training error characterization mainly follows from exiting works e.g., Fallah et al., 2020, and the stability analysis is somewhat based on Bousquet \& Elisseeff, 2020. Can the authors clarify here what is challenging when extending to the MAML training?
>
> **Response:** Thanks for your comment. We should highlight that the major contribution of our paper is its generalization bound and the established training error is stated mostly for completeness of our presentation.
>
>  Regarding our generalization bound, we should emphasize that if similar to the work by [Chen et al. 2020], we use the algorithmic stability definition of the work by [Bousquet \& Elisseeff, 2020], our generalization bound would have been $O(1/m)$. Instead of taking that path, we introduced a novel notion of algorithmic stability that better captures the essence of MAML-type methods, and as a result, it leads to a generalization bound of $O(1/mn)$.
>
>  As the reviewer has correctly pointed out, another major contribution of our analysis is eliminating the dependence on $K$ for the total variation distance characterization via the maximum coupling technique. Indeed, this approach is not considered or studied in other works on generalization of MAML-type methods.
>
> We also refer the reviewer to our response to a comment by Reviewer aLZn on the tightness of our bound for the excess error of unseen tasks.

---

> > ### Author Response · Authors · 2021-09-02
> > **Additional responses to Reviewer CKWF**
> >
> > **Comment: The authors provide a generalization bound for an unseen task, which increases with the total variation distance between tasks’ distributions. Therefore, it would be better if the authors can provide experiments to verify whether such a total variation distance metric really captures the generalization error.**
> >
> > We wanted to follow up on the reviewer's concern regarding the unseen task result and provide further explanation from two aspects:
> >
> > First, we refer the reviewer to the example provided in our response to Reviewer aLZn:
> > Consider the loss function $l(w,(x,y)) = \frac{1}{2} (w^\top x - y)^2$. In this case, it is straightforward to see that the expected loss of task $i$ is given by
> >
> >  $$ L_i(w) = E_{(x,y)\sim p_i} [l(w,(x,y))] = \frac{1}{2} w^\top \Sigma_i w - w^\top \rho_i + \sigma_i^2,$$
> >
> >  where $\Sigma_i = \mathbb{E}\_{(x,y) \sim p\_i}[x x^\top]$, $\rho\_i = \mathbb{E}\_{(x,y) \sim p\_i}[xy]$, and $\sigma_i^2 = \mathbb{E}\_{(x,y) \sim p\_i}[y y^2]$. Let us assume $\Sigma\_i = \gamma\_i I_d$ where $\gamma\_i$ is drawn from a uniform distribution over $(1-\zeta, 1+\zeta)$. Also, we assume $\rho_i$ is drawn from an independent distribution with mean $\bar{\rho}$. Let $w^*$ be the solution of the MAML problem
> >
> >  $$ \min\_{w}\frac{1}{m} \sum\_{i=1}^m L_i(w - \alpha \nabla L\_i(w)).$$
> >
> >  From reference [13] in the paper, we know that
> >
> >  $$w^* = (\frac{1}{m} \sum_{i=1}^m (I - \alpha \Sigma_i)^2 \Sigma_i)^{-1} (\frac{1}{m} \sum\_{i=1}^m (I - \alpha \Sigma_i)^2 \rho_i).$$
> >
> > We can see that for large $m$ the first term on the right hand side concentrates around
> >
> > $$\mathbb{E}\_{\Sigma}[((I - \alpha \Sigma)^2 \Sigma)^{-1}] $$
> >
> > and the second term concentrates around
> >
> > $$\mathbb{E}\_{\Sigma, \rho}[(I - \alpha \Sigma)^2 \rho].$$
> >
> > Ignoring terms of order $\mathcal{O}(\alpha^2)$, and using the fact that $\mathbb{E}[\Sigma^2] = (1 + \zeta^2/3) I_d$, we can see that for large $m$ we approximately have
> >
> > $$w^* = \frac{1 - 2\alpha}{1-2\alpha - 2/3 \alpha \zeta^2} \bar{\rho}.$$
> >
> > Now, we look at the value of the obtained solution in the objective function of the new task $m+1$. Using strong convexity, we can use the gradient norm $\mathbb{E}\_{\Sigma\_{m+1}, \rho\_{m+1}} [|| \nabla F\_{m+1}(w^*) ||^2]$ as a surrogate for the sub-optimality. Note that if we show this term is bounded below by a non-vanishing term, then we can conclude that the excess error is non-vanishing as well.
> > To show this, it suffices to lower bound $|| \mathbb{E}\_{\Sigma\_{m+1}, \rho\_{m+1}} [\nabla F\_{m+1}(w^*)] ||$. Note that
> >
> > $$ \nabla F\_{m+1}(w^*) = (I - \alpha \Sigma\_{m+1})^2 (\Sigma\_{m+1} w^* - \rho\_{m+1}),$$
> >
> > and hence for small $\alpha$ we have
> >
> > $$ || \nabla F\_{m+1}(w^*) || \geq (1 - \alpha (1+\zeta))^2  || \Sigma\_{m+1} w^* - \rho\_{m+1} ||.$$
> >
> > This implies that
> >
> > $$ \mathbb{E} [ || \nabla F\_{m+1}(w^*) || ] \geq (1 - \alpha (1+\zeta))^2 \mathbb{E} [ || \Sigma\_{m+1} w^* - \rho\_{m+1} ||]$$
> > $$ \geq (1 - \alpha (1+\zeta))^2 || \mathbb{E}[\Sigma\_{m+1} w^* - \rho\_{m+1}] || $$
> > $$ = (1 - \alpha (1+\zeta))^2 || w^* - \bar{\rho} ||$$
> > $$ ~= (1 - \alpha (1+\zeta))^2 (\frac{1 - 2\alpha}{1-2\alpha - 2/3 \alpha \zeta^2}-1) ||\bar{\rho}||$$
> >
> > As the above lower bound shows, if the parameter  $\zeta$ is zero, then the lower bound becomes zero. However, for the case that $\zeta$ is non-zero (which means that the second moment of their input data points $x$ are different from each other), then the lower bound is strictly larger than zero, and hence, one can not hope for a zero error for the new task. It is worth noting that, as shown by [1], when the distribution of $x$ is Gaussian, the TV distance between distributions is $\Omega(\min(1,\sqrt{d}\zeta))$ in our example, showing the relation between TV distance and error.
> >
> > Next, we verify this point by some numerical experiments (This setting is very similar to the setup in Appendix G of the paper). Suppose the data points of each task are generated according to a Gaussian distribution, where the covariance matrix is a fixed matrix for all tasks, and the mean for task $i$ is $a_i$ defined as $a_i = \frac{u_i + \eta 1_d}{|| u_i + \eta 1_d ||}$, where $u_i$ is a random Normal vector. Note that if the covariance of feature vectors for all tasks are the same, as considered here, then the parameter $\zeta$ captures the gap between the mean of feature vectors of different tasks. In this case, by increasing $\eta$,  we are making $a_i$'s closer to each other, and, therefore, decreasing the value of $\zeta$.
> >
> >
> > Here we provide the average error of the trained model on a new task for different choices of $\eta$. The average is computed based on 40 different realizations.
> >
> >
> > | $\eta$      | error |
> > | ----------- | ----------- |
> > | 0      | 0.11       |
> > | 0.5   | 0.055     |
> > | 1      | 0.036      |
> > | 2      | 0.021       |
> >
> > As we observe, by increasing $\eta$ and decreasing $\zeta$, the error over the new task decreases which matches the explanation that we provided above.
> >
> > While we acknowledge that the explanation above is not rigorous enough and the experiment might be just a toy example, we just want to use this space to provide a short elaboration on why we believe our result sheds light into the generalization of MAML to unseen tasks. We will include the rigorous calculation of the above example as well as a more comprehensive version of the aforementioned experiment in our revised paper.
> >
> > [1]: Devroye, Luc, Abbas Mehrabian, and Tommy Reddad. "The total variation distance between high-dimensional Gaussians." arXiv preprint arXiv:1810.08693 (2018).

---

> > > ### Comment · Reviewer_CKWF · 2021-09-04
> > > **Thanks for further response**
> > >
> > > Dear Authors,
> > >
> > > Many thanks for your further justifications and the additional experiments, and I increase my score accordingly.
> > >
> > > Best,
> > > Reviewer

---

### Official Review · Reviewer_UMVa · 2021-07-19

**Rating:** 6
**Confidence:** 4

**Summary:**

The paper provides a generalization analysis of MAML. Based on a new definition of stability of meta-learning, it characterizes the generalization error on seen and unseen tasks.


**Limitations And Societal Impact:**

The assumption of strong convexity and smoothness is very strong and does not hold for most neural networks. The authors have not fully addressed such limitation.


**Main Review:**

Please address the following concerns:
- The results in Theorem 2 and Corollary 2 both show that the generalization error increases for a larger K (the number of shots). This conflicts with the experiment results in MAML or other meta-learning papers. So I have a major concern about whether the bound is vacuous. Moreover, it would be better to empirically verify the effect of K.
- The theoretic results of Theorem 2, i.e., the generalization error of seen tasks, are not empirically studied. It would also better to provide some simple experiments to verify it.
- The equation in line 569 seems not so straightforward. The loss of the i-th task is not only affected by the replacement of K data points in $S_i^n$, but also affected by the replacement of data points in $S_j^n (j \neq i)$. So I wonder whether expectations of $w_S$ and $w_\tilde{S}$ are the same.

****************************************************After Rebuttal**********************************************
- While I am satisfied with the response to the assumption of strong convexity and smoothness, I am still not convinced by the response to my major concern regarding $K$.
     - When $K$ is small which is usually the case in meta-learning, this bound is as loose as SGD. In this case, why do we need meta-learning?
     - Only when $K$ is as large as possible (Appendix E), the error decreases as $K$ increases. In fact, according to the bound, if the total number of samples across all tasks is $1M$, $K$ has to be even as large as $\sim10,000$, which is rarely practical in meta-learning.

If the latest discussion by the authors can be included, I tend to increase the score to "weakly accept".

**Time Spent Reviewing:**

5

---

> ### Author Response · Authors · 2021-08-11
> **Response to Reviewer UMVa**
>
> We thank the reviewer for the constructive feedback. In the following paragraphs we briefly address the issues raised by the reviewer.
>
> **Comment:** The results in Theorem 2 and Corollary 2 both show that the generalization error increases for a larger $K$ (the number of shots). This conflicts with the experiment results in MAML or other meta-learning papers. So I have a major concern about whether the bound is vacuous. Moreover, it would be better to empirically verify the effect of $K$.
>
> **Response:** We do not believe our bound is vacuous or trivial. The classic lower bound for SGD over strongly convex functions translates to a $\mathcal{O}(1/mn)$ lower bound in our setting, which matches the bound that we have established for the regime that $K$ is small. Moreover, our $\mathcal{O}(1/mn)$ generalization bound improves the best-known bound for MAML-type methods which is
> $\mathcal{O}(1/m)$. Therefore, our generalization bound is tight for the case that $K$ is small (relative to $n$). We should add that this setting is often the case for most real-world applications of  meta-learning, such as $K$-shot learning. In addition, we provide a different bound in Appendix E which is tighter than Theorem 2's bound for the large $K$ regime. We will highlight this point in the revised paper.
>
> $~$
>
> **Comment:** The theoretic results of Theorem 2, i.e., the generalization error of seen tasks, are not empirically studied. It would also better to provide some simple experiments to verify it.
>
> **Response:**
> Please see Appendix G, Figure 2. In this figure, we illustrate the test error over recurring tasks for a linear regression problem.
>
>
> $~$
>
> **Comment:** The equation in line 569 seems not so straightforward. The loss of the i-th task is not only affected by the replacement of K data points in $S_i^n$, but also affected by the replacement of data points in $S_j^n (j \neq i)$. So I wonder whether expectations of $w_S$ and $w_{\tilde{S}}$ are the same.
>
>
> **Response:**  Note that our definition of stability says that we only replace the data points of $\mathcal{S}\_{i}^{in}$ and $\mathcal{S}\_{i}^{out}$ for one $i$. In other words, we do not change the data points of more than one task, i.e., the data points for $j\neq i$ stays unchanged.
>
> That said, to show Theorem 1, in line 559, we say that it suffices to show
>
> $$\mathbb{E}_{A,S}[F_i(w_S) - \hat{F}_i(w_S, S_i)] \leq \gamma,$$
>
> for any $i$. To show this for a fixed $i$, we use the definition of stability by only changing $\mathcal{S}\_{i}^{in}$ and $\mathcal{S}\_{i}^{out}$.
>
> We hope this explanation addresses your question. Please let us know if further clarification is needed for this proof.
>
>
> $~$
>
> **Comment:** The assumption of strong convexity and smoothness is very strong and does not hold for most neural networks. The authors have not fully addressed such limitation.
>
> **Response:**
> We agree with the reviewer that our assumptions do not hold for neural networks in general nonconvex functions. In fact, we have discussed this limitation in Section 4 of the paper. However, as we have stated in Remark 2, the main reason that we have imposed the strongly convex assumption is that the algorithmic stability technique for deriving generalization bounds is mainly limited to convex settings and its application to nonconvex settings leads to suboptimal training error. Next, we briefly explain this issue.
>
> Consider Theorem 3.12 in Section 3.5 of [Hardt et al., 2016] where a generalization bound using algorithmic stability technique is provided for nonconvex settings. There, the authors assume the stepsize $\alpha_t$ satisfies the condition $\alpha_t \leq c/t$. To see how this prohibits us from finding a first-order stationary point (FOSP) efficiently, let us recall the convergence analysis of a non-convex smooth objective function $f$. There the main inequality is the following (see Section 1.2.3 in [33]):
>
> $$f(w_T) - f^* \geq \sum_{t=0}^T \alpha_t (1-\alpha_t L/2) \| \nabla f(w_t) \|^2,$$
> where $L$ is the smoothness parameter and $w_t$ is $t$-th iterate. It can be shown that by setting the stepsize to $\alpha_t = \Theta(1/t)$, as suggested by [Hardt et al., 2016], we would require  $\exp(\Theta(1/\epsilon^2))$ iterations to find an $\epsilon$-FOSP. However, with a constant stepsize, we can achieve the significantly improved rate of $\mathcal{O}(1/\epsilon^2)$ which matches the lower bound for this setting. As this argument shows, to obtain a meaningful generalization bound using algorithmic stability the stepsize should be selected much smaller than the required threshold and as a result the overall iteration/sample complexity could be very large.
>
> Considering this discussion, the algorithmic stability technique imposes a very restrictive assumption on the stepsizes in the *nonconvex setting* which has a detrimental effect on the training error analysis.

---

> > ### Author Response · Authors · 2021-08-31
> > **Response to Further Comments by Reviewer UMVa**
> >
> > We thank the reviewer for additional comments and feedback. Regarding your question:
> >
> > - This is true that for small $K$ our generalization bound is similar to SGD. In fact, we should not expect anything better as SGD is a special case of MAML with $\alpha=0$. However, this does not mean that we do not gain over SGD by using MAML. Note that the generalization error and the training error bound the performance of the model with respect to the baseline (e.g., $\min_{w \in \mathcal{W}} F$ for MAML), but these baselines are not the same for MAML and SGD. In fact, the gain is hidden in the baseline: For MAML the baseline is the minimum loss after one update, i.e, $\min F(.)$, while for SGD the baseline is the minimum of average losses without any further update which is larger than the MAML baseline.
> >
> > - For large $K$, while we are not sure whether our bound is tight, it is worth noting that this is not necessarily the case that the generalization bound improves as $K$ increases. to see this, consider MAML with only one task, i.e., $m=1$, and the quadratic loss $l(w,z) = (w^\top x - y)^2$ with $z=(x,y)$. In addition, and to focus on the generalization error coming from test update, we assume we have access to exact gradients for outer loop, i.e.,
> > $$\hat{F}(w)= \frac{1}{\binom{n}{K}} \sum_{\{z_i\} \subset \mathcal{D}^{in}} \mathcal{L} \left (w - \alpha \sum_{i=1}^K \frac{1}{K} \nabla l(w,z_i) \right )$$
> > Let $\Lambda = \mathbb{E}[xx^\top]$ and $\rho = \mathbb{E}[xy]$. Also, we denote the estimation of $\Lambda$ and $\rho$ over $\mathcal{D}^{in}$ by $\hat{\Lambda}$ and $\hat{\rho}$, respectively.
> >
> > Doing the algebra, it can be seen that
> >
> > $$\nabla F(w) = \Lambda w - \rho - 2 \alpha \Lambda^2 w + 2 \alpha \Lambda \rho + \mathcal{O}(\alpha^2) $$
> >
> > $$\nabla \hat{F}(w) = \Lambda w - \rho + 2 \alpha \Lambda \hat{\Lambda} w + \alpha \Lambda \hat{\rho} + \alpha \hat{\Lambda} \rho + \mathcal{O}(\alpha^2)$$
> >
> > It can be seen that the difference of the two gradients is $\Omega(\frac{\alpha}{n})$ and does not decrease as $K$ increases (note that the $\alpha$ in bound is because of the simplifying assumption that we made on having exact gradient for outer loop).
> >
> > It is worth noting that, similar to the above discussion, this does not mean that the MAML performance does not improve as $K$ increases. Increasing $K$ makes the gradient update at test time closer to the exact gradient, and hence the baseline ($\min F$) decreases.

---

### Official Review · Reviewer_acnS · 2021-07-26

**Rating:** 6
**Confidence:** 2

**Summary:**

This paper analyzes the generalization bound of MAML-type algorithms from two aspects: (1) when the test task is a sample from the training tasks and (2) when the test task is unseen during training stage. A generalization bound on strongly convex case is given in case (1) showing the generalization error is deceasing at a rate of $O(1/mn)$. On the other hand, a bound under the total variation assumption is given in case (2), showing the bound depends on the average total variation between all the training tasks and the unseen task. The paper focuses on theory and thus no experiments are included.

**Limitations And Societal Impact:**

No limitation or potential negative societal impact is shown.

**Main Review:**

Overall I think this is a good paper, because it provides the generalization bound in terms of data points in the homogeneous case, and a total variation based analysis in the heterogeneous case. I didn't check all the proof line by line, but the key results make sense to me and are, to my best knowledge, new in the literature.

Regarding the technical parts, I have two concerns:

1. The authors state in line 292 Remark 2 that in the analysis of recurring tasks case, strong convexity is assumed because "The algorithmic stability technique is mainly limited to the convex setting." This seems a little off to me, because algorithmic stability has been studied in several previous works. For instance, Section 3.5 of (Hardt et al., 2016; https://arxiv.org/pdf/1509.01240.pdf), (London, 2017; https://arxiv.org/pdf/1709.06617.pdf), (Zhang et al., 2020; https://arxiv.org/pdf/2002.11875.pdf), etc. It seems the divergence comes from a different notion of stability proposed in the paper, which captures the number of data points per task. But I don't see how such definition would prevent the analysis on non-convex problems. Could you elaborate why the non-convexity assumption is challenging based on the stability notion in your case? Maybe an example would help.

2. A minor concern: In many previous works analyzing MAML-type algorithms, a bilevel structure is established (which is closer to how we do meta-learning in reality). For instance (Ji et al., 2020;https://arxiv.org/pdf/2006.09486.pdf) analyzes the convergence, etc. I wonder how the results in this paper would adapt/be applicable to the bilevel case? In bilevel programming, the stability analysis is not new(https://arxiv.org/pdf/2106.04188.pdf), do you see any challenge there?

**Updates**
I've read the author response and I think the authors adequately answered my question regarding difficulty in relaxing the results to the non-convex case. I encourage the authors to include the discussion into the paper so as to make it clear.

**Time Spent Reviewing:**

5

---

> ### Author Response · Authors · 2021-08-11
> **Response to Reviewer acnS**
>
> We thank the reviewer for the constructive feedback. In the following paragraphs we briefly address the issues raised by the reviewer.
>
> **Comment:**
> The authors state in line 292 Remark 2 that in the analysis of recurring tasks case, strong convexity is assumed because "The algorithmic stability technique is mainly limited to the convex setting." This seems a little off to me, because algorithmic stability has been studied in several previous works. For instance, Section 3.5 of (Hardt et al., 2016; https://arxiv.org/pdf/1509.01240.pdf), (London, 2017; https://arxiv.org/pdf/1709.06617.pdf), (Zhang et al., 2020; https://arxiv.org/pdf/2002.11875.pdf), etc. It seems the divergence comes from a different notion of stability proposed in the paper, which captures the number of data points per task. But I don't see how such definition would prevent the analysis on non-convex problems. Could you elaborate why the non-convexity assumption is challenging based on the stability notion in your case? Maybe an example would help.
>
> **Response:**
> We thank the reviewer for asking this excellent question. The main issue with applying algorithmic stability techniques for the non-convex case is that we have to select a very small stepsize to obtain reasonable generalization bounds, but with such small stepsizes, we cannot guarantee that we will find a first order stationary point (FOSP) solution of the empirical loss in polynomial time. We'd like to mention all previous works that use of algorithmic stability for deriving generalization bounds in nonconvex settings suffer from this issue. Next, we provide more details about this point.
>
> To be more precise, consider Theorem 3.12 in Section 3.5 of [Hardt et al., 2016]. There, the authors assume the stepsize $\alpha_t$ satisfies the condition $\alpha_t \leq c/t$. To see how this prohibits us from finding an FOSP efficiently, let us recall the convergence analysis of a non-convex smooth objective function $f$. There the main inequality is the following (see Section 1.2.3 in [33]):
>
> $$f(w_T) - f^* \geq \sum_{t=0}^T \alpha_t (1-\alpha_t L/2) \| \nabla f(w_t) \|^2,$$
> where $L$ is the smoothness parameter and $w_t$ is $t$-th iterate. It can be shown that by setting the stepsize to $\alpha_t = \Theta(1/t)$, as suggested by [Hardt et al., 2016], we would require  $\exp(\Theta(1/\epsilon^2))$ iterations to find an $\epsilon$-FOSP. However, with a constant stepsize, we can achieve the significantly improved rate of $\mathcal{O}(1/\epsilon^2)$ which matches the lower bound for this setting. As this argument shows, to obtain a meaningful generalization bound using algorithmic stability the stepsize should be selected much smaller than the required threshold and as a result the overall iteration/sample complexity could be very large.
>
> Considering this discussion, the algorithmic stability technique imposes a very restrictive assumption on the stepsizes in the *nonconvex setting* which has a detrimental effect on the training error analysis.
>
> Regarding the other papers mentioned by the reviewer, the work in (London, 2017) uses the result in [Hardt et al., 2016] for the nonconvex case, and hence, it suffers from the same issue. Finally, the work by (Zhang et al., 2020) studies a different notion of stability and their focus is not on algorithmic stability or generalization of algorithms.
>
> $~$
>
> **Comment:**
> A minor concern: In many previous works analyzing MAML-type algorithms, a bilevel structure is established (which is closer to how we do meta-learning in reality). For instance (Ji et al., 2020;https://arxiv.org/pdf/2006.09486.pdf) analyzes the convergence, etc. I wonder how the results in this paper would adapt/be applicable to the bilevel case? In bilevel programming, the stability analysis is not new(https://arxiv.org/pdf/2106.04188.pdf), do you see any challenge there?
>
> **Response:**
> We believe our analytical techniques and our stability definition can be extended to the bilevel optimization setting as long as the training and validation sets are selected separately (which is often the case [Bai et al., 2021]). Indeed, this is a valid research problem to investigate, but we believe it is beyond the scope of this paper.
> Finally, we would like to emphasize that the second paper which utilizes stability analysis for bilevel programming has been posted on arXiv after the NeurIPS deadline, and hence, we believe it is not fair to be considered in the assessment of our paper. However, we will make sure to discuss it in the final version of our manuscript.
>
> [Bai et al., 2021] Bai, Y., Chen, M., Zhou, P., Zhao, T., Lee, J., Kakade, S.,Wang, H., and Xiong, C. (2021). How important is the train-validation splitin meta-learning?  InProceedings of the 38th International Conference onMachine Learning, volume 139 ofProceedings of Machine Learning Research,pages 543–553. PMLR.

---

### Decision · Program_Chairs · 2021-09-27

**Decision:**

Accept (Poster)

**Comment:**

This learning-theory paper used stability approach to study the generalization properties of Model-Agnostic Meta-Learning (MAML).  The rate of $O(1/mn)$ for strongly convex objective in the homogeneous setting is the first-ever known result for meta learning and greatly improves the existing results.  For the heterogeneous setting, the generalization was proved using the total variation distance between the data distributions of this unseen task and the training tasks.  As far as I can see from reading through the proof techniques, it is indeed solid and non-trivial.

There are various concerns about strong-convexity and smoothness assumption for stability analysis which were well responded and explained in the rebuttal.  It should be included as a discussion on the limitation of stability analysis in the final version.  I would also particularly encourage the authors to include discussions on other concerns raised by the reviewers such as the generalization bound related to K and the appropriateness of using the TV metric for deriving generalization on unseen tasks for meta learning etc.